# Decoding the spatial chromatin organization and dynamic epigenetic landscapes of macrophage cells during differentiation and immune activation

Immunocytes dynamically reprogram their gene expression profiles during differentiation and immunoresponse. However, the underlying mechanism remains elusive. Here, we develop a single-cell Hi-C method and systematically delineate the 3D genome and dynamic epigenetic atlas of macrophages during these processes. We propose "degree of disorder" to measure genome organizational patterns inside topologically-associated domains, which is correlated with the chromatin epigenetic states, gene expression, and chromatin structure variability in individual cells. Furthermore, we identify that NF-κB initiates systematic chromatin conformation reorganization upon *Mycobacterium tuberculosis* infection. The integrated Hi-C, eQTL, and GWAS analysis depicts the atlas of the long-range target genes of mycobacterial disease susceptible loci. Among these, the SNP rs1873613 is located in the anchor of a dynamic chromatin loop with *LRRK2*, whose inhibitor AdoCbl could be an anti-tuberculosis drug candidate. Our study provides comprehensive resources for the 3D genome structure of immunocytes and sheds insights into the order of genome organization and the coordinated gene transcription during immunoresponse.

The distinct and dynamic epigenetic codes of different cell types function as blueprints for precise spatial and temporal gene transcription in different developmental stages in response to different stimuli, such as immune cells during differentiation and immunological responses[1,2]. DNA epigenetic codes are mainly embedded in various types of DNA and histone protein modifications[3], chromatin accessibility[4], and in the sophisticated organization of the three-dimensional (3D) genome[5]. Epigenetic modifications in enhancers, which are often elegantly looped with target promoters in a topologically associating domain (TAD), can reprogram the transcription of cell-type-specific genes[6]. While it was historically believed that TADs are conserved across cell types[7], emerging evidence suggests that TADs undergo dynamic changes during differentiation, especially at single-cell level[8,9]. However, the mechanisms by which the

four-dimensional (4D; 3D plus time dynamics) genome architecture and the integrated epigenetic codes orchestrate transcriptional programs during diverse differentiation and physiological processes, especially in immunological responses, are poorly characterized.

Macrophages are differentiated from monocytes and play a crucial role in the immunological defense against invading pathogens[10]. During differentiation, the staining of the monocytes nucleus, which is originally a rounded shape, dynamically changes to a kidney bean shape or U shape[11], implying a dynamic 3D chromatin structure. The previous study has also demonstrated global changes in chromatin looping events during macrophage development[12]. Upon infection, macrophages can be polarized into different cell subtypes, such as M1 and M2 macrophages, with distinct functional and phenotypic properties. M1 macrophages can be activated by lipopolysaccharide (LPS)

e-mail: guoliang.li@mail.hzau.edu.cn; gcao@mail.hzau.edu.cn

or IFN-γ, and exhibit robust anti-microbial activity. M2 macrophages are alternatively activated by IL-4 and IL-13 and play an important role in allergic diseases, angiogenesis, and tissue remodeling[13,14].

Histone modifications of chromatin play an important role in the regulation of gene expression. For instance, the simultaneous presence of H3K4me3 and H3K27me3 chromatin (bivalent chromatin) in promoter and enhancer regions can inhibit the expression of development-related genes and regulate the development process of embryonic stem cells and cranial neural crest cells[15,16]. It has been demonstrated that histone methylation and demethylation are instrumental in regulating the expression of cytokines, chemokines, and transcription factors, and subsequently modulating macrophage polarization[17,18]. A deeper understanding of the epigenetic regulation of macrophage phenotypes is expected to facilitate the development of gene-specific therapeutic approaches that can enhance host defense while preserving tissue integrity[17].

Tuberculosis (TB) is a global infectious health threat even more lethal than HIV/AIDS, leading to approximately 1.3 million deaths and 10.4 million new cases worldwide annually[19]. Upon *Mycobacterium tuberculosis* (M.tb) invasion into the lung, pulmonary monocytes differentiate into pulmonary macrophages and are activated for immune defense against *M.tb* infection[20]. Extensive genome-wide association studies (GWASs) have identified many mycobacterial disease susceptibility loci[21–28], of which the majority are noncoding regulatory genetic elements. However, it remains unknown which genes are the distal targets of these noncoding regulatory loci and how the SNPs influence the 3D genome.

In this work, we use the plastic THP-1 monocytes and the differentiated macrophages as a model system and delineate the comprehensive spatial chromatin organization and dynamic epigenetic landscapes of macrophages during differentiation and infection with *M.tb*. Furthermore, we propose a concept of TAD "Degree of Disorder" to measure the entropy of chromatin architecture inside immune-related TADs, which is correlated with chromatin accessibility, gene expression, and co-regulation during differentiation and activation. The integrated GWAS, Hi-C, eQTL, ChIP-Seq, and ATAC-Seq analysis identify the long-range target genes of mycobacterial disease-susceptible SNPs and identify *LRRK2* as a potential drug target, whose inhibitor AdoCbl has an anti-tuberculosis effect both in vitro and in vivo. To verify the function of rs1873613, we introduce a single base pair mutation in THP-1, and verify that rs1873613 can enhance the *LRRK2* enhancer activity, and thereby upregulate the expression of the *LRRK2* gene.

## Results

### Morphology, transcriptome, and epigenetic dynamics during macrophage differentiation and activation upon *M.tb* infection

To investigate the spatial chromatin organization and epigenetic dynamics of macrophages during differentiation and immune activation, we used THP-1 cells, a highly plastic human monocytic cell line that can be differentiated into macrophage-like cells and activated by infection, as a cell model. In our study, we named the THP-1 cells before differentiation, and after differentiation, and activated by *M.tb* H37Ra as Thp1-mono, Thp1-macro, and Thp1-*M.tb,* respectively. As shown in Fig. 1a, b, the nuclei of differentiated cells were transformed from ellipsoid to kidney bean shape or U shape upon phorbol 12-myristate-13-acetate (PMA) treatment, implying a possible 3D genome conformational change during this process. The distinct morphology alternation of this cell line during different lineages might be underlying a rapid reprogramming of gene expression profile and epigenetic codes during differentiation and activation.

Thus, we analyzed the transcriptomes of Thp1-mono, Thp1-macro, and Thp1-*M.tb* (Fig. 1c, d and Supplementary Fig. 1a−c), and identified the differentially expressed genes (Supplementary Data 1, 2). During differentiation from monocytes to macrophages, the expression levels

of a group of genes related to development and differentiation, such as *HOXs*, *BCL11A*, *MYC*, and *NRG1*, were significantly altered (Fig. 1c). After infection, the expression levels of immune-related chemokine and transcription factors[29,30], such as *NFKB1*, *GBP1*, *CCL2*, and *IFIT2*, were significantly increased (Supplementary Data 2). In addition, M1 macrophage marker genes such as *CD80, CCR7, INHBA,* and *TNF-a*[31,32] were significantly upregulated (Fig. 1d and Supplementary Data 2), indicating that the macrophages are activated and transformed to the M1 phenotype. Interestingly, the programmed death ligand 1 (*PD-L1*) gene, which can counteract activated T-cells, was dramatically upregulated after *M.tb* infection (Fig. 1d). Gene Ontology (GO) enrichment analysis also revealed that the upregulated genes were significantly enriched in innate immune biological processes related to the response to external stimulus, defense response, and immune response (Supplementary Fig. 1c).

To delineate the dynamic epigenetic atlas of this cell during differentiation and activation, we systematically investigated the changes in chromatin states by the assay for transposase-accessible chromatin with sequencing (ATAC-Seq) and comprehensive histone chromatin immunoprecipitation and sequencing (ChIP-Seq). The modifications investigated included H3K27me3, H3K9me3, H3K4me3, H3K27ac, and H3K4me1. By combining these epigenetic modification data, we classified the chromatin states of Thp1-mono, Thp1-macro, and Thp1-*M.tb* cells into a 15-state model comprising 8 active states and 7 repressed states via the ChromHMM method[33] (Fig. 1e). As shown in Fig. 1e, the chromatin regions enriched with H3K4me3 and H3K27ac showed relatively higher gene expression levels than unenriched regions. While the overall chromatin states of the whole genome were not dramatically altered, we observed dynamic changes in chromatin states in several chromosome regions, such as bivalent and genic enhancers (Fig. 1e).

To further investigate the dynamics of the enhancer and promoter epigenetic states, we comprehensively combined the data for H3K4me1, H3K4me3, H3K27ac, and H3K27me3 modification enrichment with ATAC-Seq peaks and defined the states of enhancers and promoters according to methods described in the previous studies[3,16,34]. Supplementary Fig. 1d, e illustrates primed enhancers, poised enhancers, active enhancers, repressed promoters, bivalent promoters, and active promoters. The genes marked by active promoters had the highest expression levels in the RNA-Seq data, whereas those marked by repressed promoters had the lowest expression levels (Supplementary Fig. 1f). These results support the integrity of our histone modification and ATAC-Seq data, as well as our analysis of promoter and enhancer states.

Next, we analyzed the dynamics of enhancer and promoter states across Thp1-mono, Thp1-macro, and Thp1-*M.tb* cells during differentiation and infection (Fig. 1f and Supplementary Fig. 1g and Supplementary Data 3). Our data demonstrated that a dynamic transition in the promoter epigenetic state, especially the transition from a repressive to an active state and vice versa, could be associated with the gene expression level (Supplementary Fig. 1h). Upon *M.tb* infection, 424 promoters changed from the repressive state to the active state (Fig. 1f and Supplementary Data 3). Kyoto Encyclopedia of Genes and Genomes (KEGG) pathway enrichment analysis showed that the genes harboring these promoters were enriched in innate immune pathways such as ligand-dependent caspase activation (24.0%), NOTCH signaling (10.7%), and NF-kappa B signaling (9.3%) (Fig. 1g). These results suggest that *M.tb* infection can systematically turn on immune-defense gene expression by reprogramming the epigenetic state of promoters and enhancers and switching macrophages to the active M1 state. For example, the promoter region of TNFSF10, a typical death ligand involved in immune surveillance[35], was much more accessible and enriched with H3K4me3 modifications after *M.tb* infection. Accordingly, its transcription was also dramatically increased (Fig. 1h).

## Dynamic chromatin architectures of THP-1 cells during differentiation and immunological response

As the nuclei of THP-1 cells were transformed from ellipsoid to kidney-shaped after differentiation and activation, the 3D genome conformation might undergo a transformation during these processes. To test this hypothesis, we generated high-resolution (~5 Kb) genome-wide chromatin interaction maps using the in situ digestion-ligation-only (DLO) Hi-C method[36,37], with ~1.5 billion sequencing reads for each library (Supplementary Table 1). Figure 2a shows a high-quality Hi-C heatmap with low noise, clearly displaying the TAD domains. With this high-resolution genome contact matrix, we identified the dynamics of the chromatin structure during macrophage differentiation and activation. For example, several strong chromatin interactions appeared around the *MYC* gene in the Hi-C contact matrix after differentiation, suggesting that

reprogramming of the chromatin configuration and the epigenetic code in the *MYC* regulatory repressor region[38] are involved in the regulation of *MYC* gene transcription (Fig. 2a and Supplementary Fig. 2a, b).

During THP-1 differentiation, 13.0% of TADs shifted at least one of their boundaries by >80 Kb, 13.8% fused into larger TADs, and 4.0% divided into small TADs. After *M.tb* infection, 1.4% fused into larger TADs, 16.2% separated into small TADs, and 10.0% shifted at least one of their boundaries (Fig. 2b). For example, before *M.tb* infection, both the *GLS* and *STAT1* genes were located in the same TAD, while after immune activation, the *STAT1* gene was relocated into a newly emerged independent TAD. The ATAC-Seq data indicated that the boundary region of this new TAD was significantly opened upon *M.tb* infection, implying that some regulatory proteins might bind in this region and thus possibly activate *STAT1* expression (Fig. 2c). The

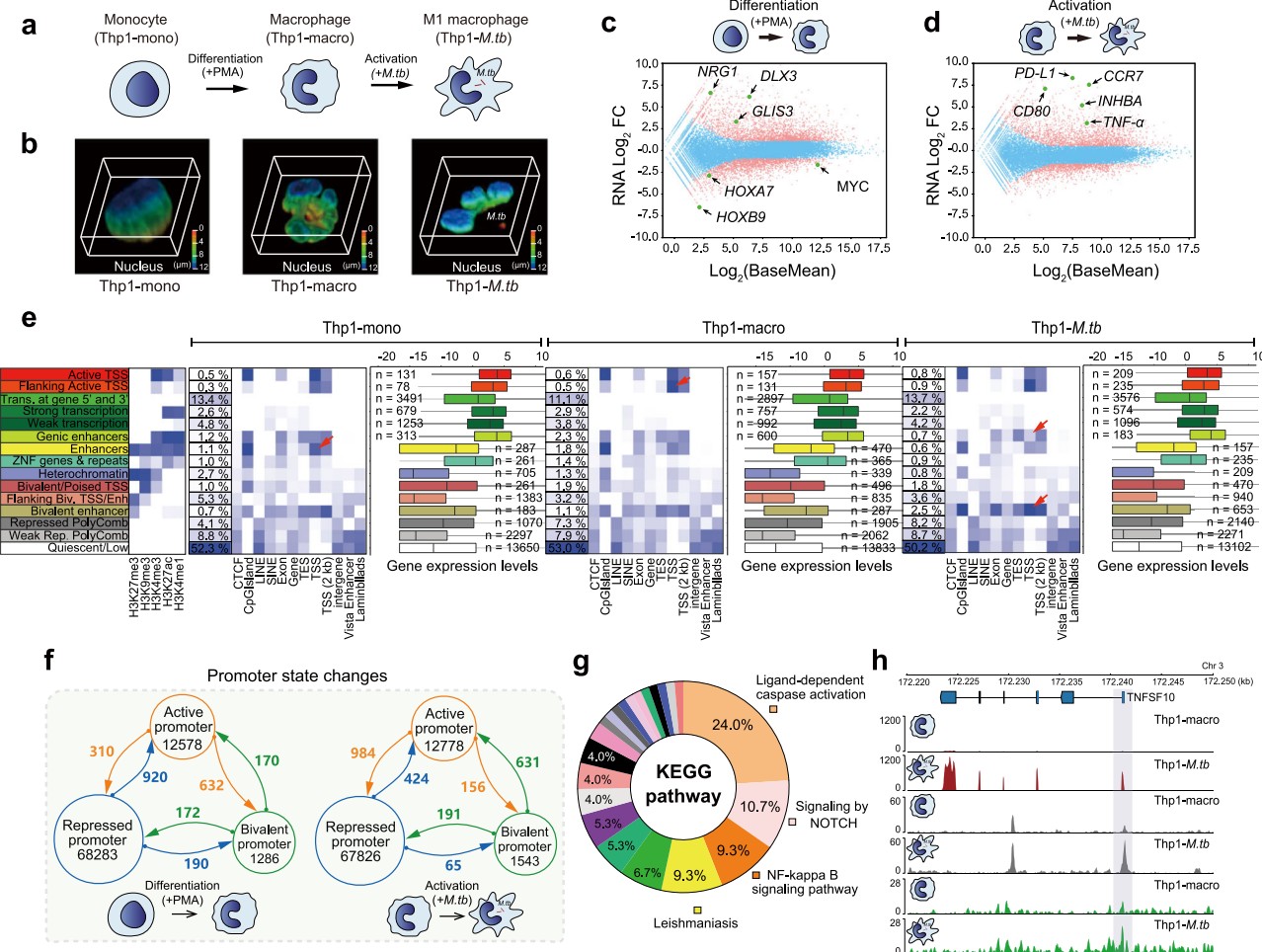

**Fig. 1 | Morphological and chromatin state dynamics of THP-1 cells during differentiation and *M.tb* infection. a** Cartoon of Thp1-mono, Thp1-macro, and Thp1-*M.tb*. **b** Reconstruction of the three-dimensional structure of the nucleus. The nuclei were stained with Hochest and reconstructed. The different colors indicate different depths. **c** MA plot for gene differential expression analysis during Thp-1 cell differentiation. *X*-axis represents the mean of normalized counts, and *Y*-axis represents the log₂ fold changes of gene expression level. Important transcription factors related to cell proliferation and differentiation are highlighted. **d** MA plot for gene differential expression analysis during *M.tb* infection. *X*-axis represents the mean of normalized counts, and *Y*-axis represents the log₂ fold changes of gene expression level. Marker genes and important transcription factors for macrophages and M1 macrophages are highlighted. **e** Chromatin state definitions and histone mark probabilities, average genome coverage, genomic annotation enrichment levels, and gene expression levels in each chromatin state of Thp1-

mono, Thp1-macro, and Thp1-*M.tb* cells. Box-plot showing the log₂ (FPKM) of genes in each chromatin state. Box-plot with midline = median, box limits = Q1 (25th percentile)/Q3 (75th percentile), whiskers = minimum and maximum values, points = outliers (>1.5 interquartile range). The sample sizes (n) in each chromatin state are labeled in the figure. The difference in the chromatin state in each sample was highlighted with an arrow. **f** Dynamics of promoter states during differentiation and *M.tb* infection. The arrows indicate the chromatin states changed from one state (tails of the arrows) to another state (heads of the arrows). **g** KEGG pathway analysis of the 423 genes whose promoter status changed from repressive to active after *M.tb* infection. KEGG analysis used the ClueGO plug-in in Cytoscape software (Version: 2.5.8). **h** Genome Browser view of gene expression levels, chromatin accessibility and histone modifications at the *TNFSF10* gene region in Thp1-macro and Thp1-*M.tb* cells.

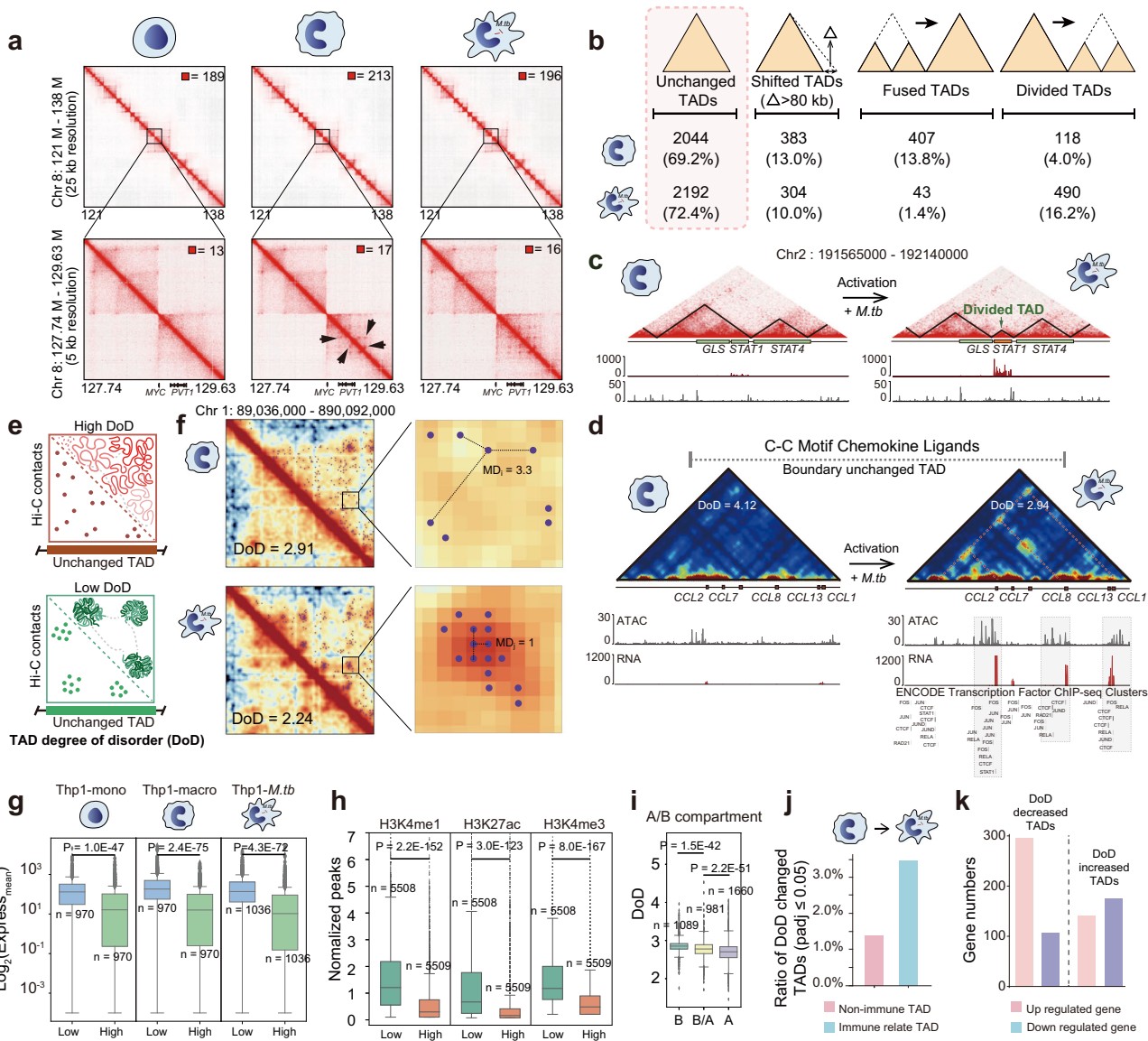

**Fig. 2 | DoD (degree of disorder) of innate immune-related TADs during differentiation and activation. a** Hi-C heatmaps at different resolutions. Strong interactions (marked with arrows) were observed around the *MYC* gene region after differentiation. **b** TAD dynamics of THP-1 cells during differentiation and activation. **c** TAD rearrangement, gene expression, and ATAC-Seq peak around *STAT1* during *M.tb* infection. **d** Typical example of CC chemokine gene cluster. The putative transcription factor (from ENCODE) were labeled with gray box. **e** The degree of disorder (DoD) was used to evaluate the organizational order of the chromatin architecture in TADs. **f** Examples of TADs with high and low DoDs and a schematic showing the DoD calculation approach. The *P*-value for observing interaction frequency is calculated based on the Poisson process. Interactions with *P*-value ≤ 0.05 are retained. **g** Relationship between TAD DoD and average gene expression level (normalized read count). The DoD value higher than median was defined as "High DoD"; lower than the median was defined as "Low DoD". Box-plot with midline = median, box limits = Q1 (25th percentile)/Q3 (75th percentile), whiskers = minimum

and maximum values, points = outliers (>1.5 interquartile range). The sample sizes (n) are labeled in the figure. *P*-values were calculated by two-side Kolmogorov–Smirnov test. **h** Correlation between TAD DoD and active chromatin epigenetic profile. The *y*-axis shows the normalized ChIP-seq peaks per TAD. Box-plot with midline = median, box limits = Q1 (25th percentile)/Q3 (75th percentile), whiskers = minimum and maximum values, points = outliers (>1.5 interquartile range). The sample sizes (n) are labeled in the figure. *P*-values were calculated by unpaired one-sided *t*-test. **i** Correlation between the TAD DoD and A, B compartments. In *x*-axis, B/A means the TADs which contains both A and B compartments. Box-plot with midline = median, box limits = Q1 (25th percentile)/Q3 (75th percentile), whiskers = minimum and maximum values, points = outliers (>1.5 interquartile range). The sample sizes (n) are labeled in the figure. *P*-values were calculated by unpaired one-sided *t*-test. **j** Compare the ratio of change of DoD value in the process of *M.tb* infection. **k** Numbers of up- and downregulated genes in the DoD decreased and increased TADs during *M.tb* infection.

KEGG analysis results demonstrated that the genes within these altered TAD boundaries during differentiation and activation were enriched in metabolic pathways and innate immune system pathways, respectively (Supplementary Fig. 2c, d). Collectively, these data suggested that the chromatin configuration was remodeled at the TAD level during differentiation and activation, which might facilitate the regulation of immunological defense-related gene expression.

## TAD "degree of disorder" of THP-1 cells during differentiation and immunological response

While most TAD boundaries remained intact (69.2%-72.4%), we found a portion of the chromatin interaction spot pattern inside TADs, especially the immune-related TADs (The TAD with immune genes located) were significantly altered, such as CC cytokine gene cluster (Fig. 2d) and *HERC* gene cluster (Supplementary Fig. 2e). Such highly organized interaction spots and randomly distributed spots might

represent different "degree of disorder" (DoD) of the chromatin architecture in the TADs (Fig. 2e). We developed an algorithm to quantify the DoD, as shown in Fig. 2f (more details are shown in Supplementary Fig. 2f–h and Methods). Briefly, we retained all the significant contact spots in the Hi-C matrix to filter the stochastic genome loci with random chromatin interactions. As the distance between the spots might reflect the similarity of the chromatin folding patterns, the overall mean distance between the spots was then calculated to represent the whole DoD value in the TAD (Fig. 2f and Supplementary Fig. 2f–h). As shown in Supplementary Fig. 2i, the DoD could indeed reflect highly organized chromatin folding patterns.

Interestingly, the average gene expression levels within TADs were negatively correlated with their DoDs (Fig. 2g). Consistent with this result, TADs structure has been demonstrated to be related to gene transcriptional regulation[39]. In addition, we found that the enhancer and active chromatin marks, such as H3K4me1, H3K27ac, and H3K4me3, were more enriched in the TADs with low DoD (Fig. 2h). In contrast, TADs with high DoD contains more transcription repression signals and heterochromatin signals (Supplementary Fig. 3a). Consistently, the A compartments, representing the transcriptionally active regions, have lower DoD than the B compartments (Fig. 2i).

Our data implied that the organization of TADs with low DoDs was more sophisticated than that of the TADs with higher DoDs. This characteristic might be mediated by the elegant cooperation with greater numbers of transcriptional regulatory factors within TADs, which may explain why the genes within the TADs with lower DoDs tend to be highly transcribed. Thus, it would be expected that the ATAC peaks in the TADs with lower DoDs should be more enriched than that of the higher ones. As shown in Supplementary Fig. 3b, the TAD DoDs was highly negatively correlated with ATAC signals within the TADs, suggesting the TAD DoD is likely associated with chromatin accessibility and the subsequent binding of transcriptional regulatory proteins.

In this scenario, the genes in highly organized TADs may be more likely to be synchronously co-regulated due to cooperation within the same transcriptional regulatory complex. To test this hypothesis, we defined the average gene co-regulation score (CRS, which means the genes synchronously upregulated or downregulated) (Supplementary Fig. 3c and Methods) within a TAD during differentiation and activation. Notably, we found that TADs with lower DoDs had overall higher CRSs (Supplementary Fig. 3d), supporting that highly organized chromatin tends to be more synchronously (or less randomly) co-regulated. As the degree of disorder is the indication of entropy, here we hypothesized that the TAD DoD might be associated with the entropy of the chromatin organization in a TAD: highly organized TADs with lower "TAD entropy" levels have overall higher gene transcription levels and more concerted co-regulation, whereas disordered TADs with higher TAD entropy levels generally exhibit lower transcription and more random co-regulation (Supplementary Fig. 3e).

Next, we analyzed the DoD dynamics of the TADs without boundary alterations during differentiation and activation. As shown in Supplementary Fig. 3f, 24.2% of TAD DoDs decreased, and 3.4% increased during cell differentiation, whereas 12.0% decreased and 5.4% increased during *M.tb* infection. Among the TADs with dynamic DoD, the overall DoD were gradually decreased when monocyte cell differentiated to macrophage cell and then to M1 cell type, which is in line with the plasticity of the cells (Supplementary Fig. 3g). Of note, our data demonstrated that the dynamic DoD, which represents the change of the chromatin organization inside the TAD, has a more profound influence on gene expression within the TAD than other types of boundary dynamics, including TAD shift, fusion, and division (Supplementary Fig. 3h). Importantly, the DoDs of

immune function-related TADs were altered more extensively than those of non-immune function-related TADs upon *M.tb* infection (Fig. 2j).

Moreover, we observed that the number of upregulated genes was larger than that of the downregulated genes when DoD was decreased, while this phenomenon did not appear during DoD increasing (Fig. 2k). For example, the TAD DoDs of CC chemokine ligand family genes were significantly decreased upon *M.tb* infection. The expression of *CCL1*, *CCL2*, *CCL7*, and *CCL8* was also synchronously upregulated (Fig. 2d). Consistent with our hypothesis, the chromatin in this TAD became more accessible after infection, which was in favor of the binding of transcriptional regulatory proteins such as NF-κB and AP1 (Fig. 2d, marked by gray boxes). Together, the alteration of TAD DoDs was underlined the reorganization of chromatin physical architecture, especially in the immune function-related TADs, and could facilitate the coordinated transcription of anti-infection genes.

## Similar chromatin folding pattern in low DoD region in individual cells

Since the bulk cell Hi-C contact matrix in the low DoD TADs is more orderly (Fig. 2d and Supplementary Fig. 2e), we speculated individual cells in the low DoD region tend to have the same or similar folding pattern and interact with a specific transcription complex, thereby forming aggregated interaction hotspots (Fig. 3a, b). To further investigate local chromatin architecture and TAD DoD at the single-cell level, we developed a single-cell-indexed DLO Hi-C (sciDLO Hi-C) to capture the chromatin conformation of individual cells based on DLO Hi-C[36] and two rounds of molecular barcoding to capture the chromatin conformation of individual cells (Fig. 3c). The advantage of this method is that only the 80 bp DNA fragments containing the proximity ligation junction were retained, which can greatly reduce the sequencing noise caused by multiple displacement amplification (Supplementary Fig. 4a–c) (see more detailed in Materials and Methods). By comparing with other single-cell Hi-C methods[9,40,41], we demonstrated that sciDLO Hi-C datasets contain the highest proportion of proximity ligation junction reads (Supplementary Fig. 4b).

By sciDLO Hi-C, we obtained 3D genome data of 409 Thp1-mono cells, 424 Thp1-macro cells, and 510 Thp1-*M.tb* cells (Supplementary Table 2). The territories between different chromosomes can be clearly distinguished based on the simulated 3D genome structure of the individual cells (Supplementary Fig. 4d, e). To investigate the heterogeneity of these three types of cells, we employed scHiCTools[42] to classify all the individual cells based on their 3D genome structures. As shown in Fig. 3d, three distinct clusters of cells were identified, representing Thp1-mono, Thp1-macro, and Thp1-*M.tb*, respectively. Among these, the Thp1-mono has obvious boundaries between the other two types of cells, whereas Thp1-macro and Thp1-*M.tb* are partially overlapped. This data reflected that PMA treatment could uniformly reprogram monocyte to a distinct cell-type macrophage, whereas macrophage activation caused by *M.tb* infection was much more heterogeneous. Notably, compared to Thp1-macro and Thp1-mono cells, Thp1-*M.tb* were enriched with significantly more chromatin contacts around the immune genes (Fig. 3e). For example, the simulated chromatin structures of individual cells showed that the chromatin interaction between innate immune-related genes *NOD2* and *BRD7* were more contacted in Thp1-*M.tb* cells (Fig. 3f, g).

To explore the order and stochasticity of the genome organization of individual cells in immune-related TADs, we further analyzed chromatin contact patterns at single-cell level. We observed that while the individual chromatin contacts displayed a certain extent of heterogeneity, the chromatin structures in low DoD regions were uniformly folded. Take the innate immune-related *STAT1* gene locus as an example (Fig. 3h–m). Upon infection, the expression of

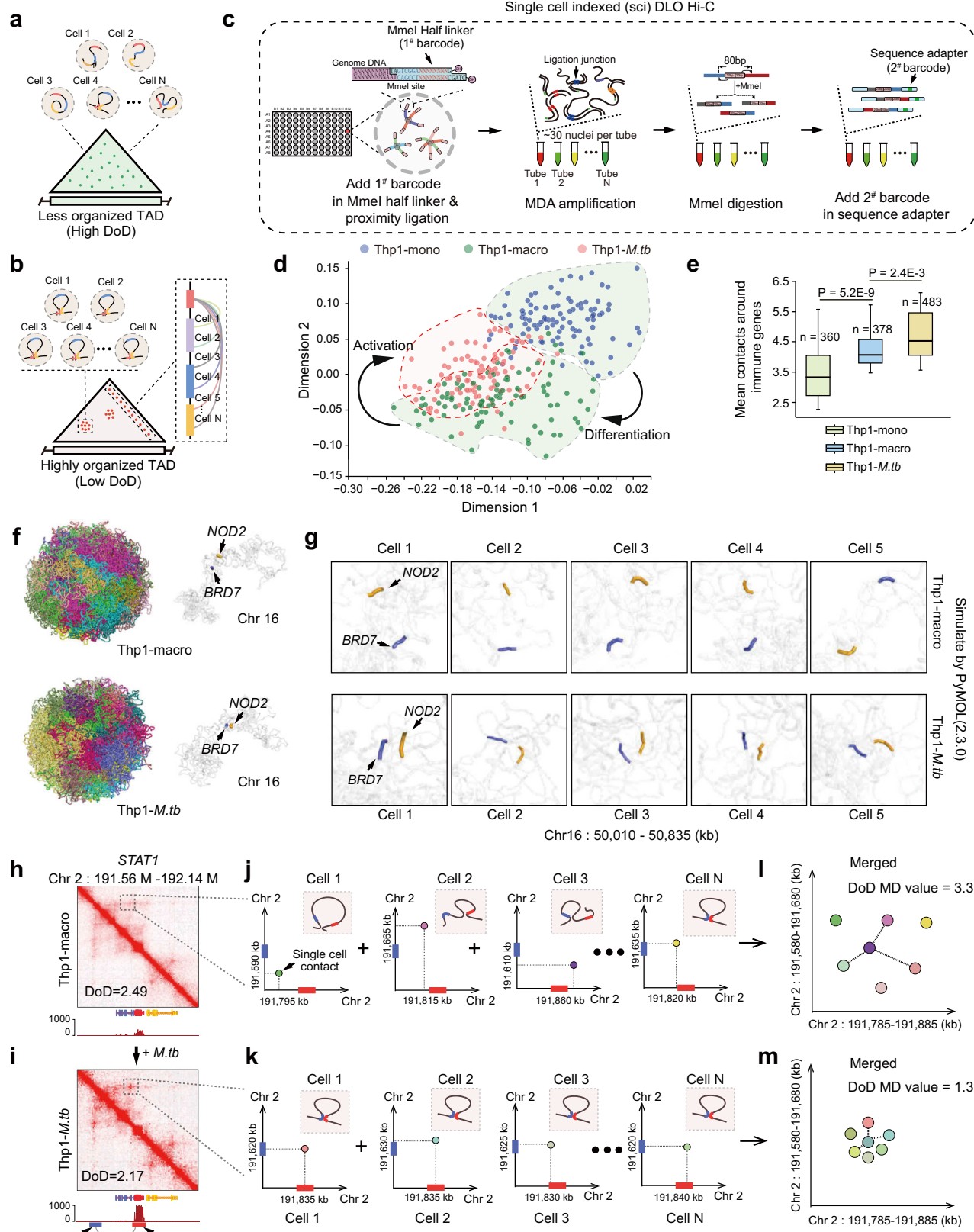

*STAT1* gene was highly upregulated, and the DoD value of *STAT1* TAD was decreased (Fig. 3h, i). We further compared the *STAT1* gene locus chromatin contact pattern between the Thp1-macro and Thp1-*M.tb* with single-cell Hi-C data (Fig. 3j–m). As shown in Fig. 3j, k, compared to the random chromatin contact pattern between *STAT1* promoter and the potential enhancer in Thp1-macro (Fig. 3j and Supplementary

Fig. 4f), the chromatin contacts of Thp1-*M.tb* in this region with relatively lower DoD values is much more consistent (Fig. 3k, m and Supplementary Fig. 4g). This data suggested that in the low DoD TADs, the individual cells tend to have similar chromatin contact pattern between different *cis*-regulatory elements, likely mediated by certain transcriptional regulatory proteins. Thus, they have an overall

**Fig. 3 | Evaluating the order and stochasticity of single-cell chromatin folding in low degree of disorder (DoD) region. a, b** Illustration of how the TAD degree of disorder (DoD) reflect the order and consistency of chromatin folding in single cells. **c** Flowchart of the single-cell indexed (sci) DLO Hi-C method. **d** Present cluster analysis result of sciDLO Hi-C datasets by using two-dimensional scatter plots. **e** Comparison of average contacts around immune genes (±10 Kb around TSS sites) between Thp1-mono, Thp1-macro, and Thp1-*M.tb* cells. Box-plot with midline = median, box limits = Q1 (25th percentile)/Q3 (75th percentile), whiskers = minimum and maximum values, points = outliers (>1.5 interquartile range). The sample sizes (n) are labeled in the figure. *P*-values were calculated by unpaired one-sided *t*-test. **f** Simulation of chromatin three-dimensional conformation of representative Thp1-macro and Thp1-*M.tb* cells by using PyMOL software (version: 2.3.0). The typical

innate immune genes *NOD2* and *BRD7* were labeled with arrows. **g** Zoom in and comparison of the spatial location of *NOD2* and *BRD7* in single-cell Thp1-macro and Thp1-*M.tb* by using simulated nucleus. *NOD2* and *BRD7* were marked with arrows. **h, i** Bulk cell chromatin contact matrix and gene expression level of *STAT1* TAD of Thp1-macro and Thp1-*M.tb* cells. The chromatin interaction hot spot which formed in low DOD TAD were marked by dashed box. Chromatin loop-mediated *STAT1* and enhancer interaction were labeled by arrows. **j, k** Single-cell chromatin contacts around *STAT1* gene of Thp1-macro and Thp1-*M.tb* cells. Individual cells have similar chromatin folding patterns in the low DoD TAD. **l, m** Calculate DOD MD (mean distance) value by using combined single-cell Hi-C contacts of Thp1-macro and Thp1-*M.tb* cells. Different colors indicate that the chromatin contacts come from different cells.

higher gene expression level, compared with the stochastically organized TADs.

## Remodeling of the chromatin configuration around GBP family genes orchestrates their coexpression upon *M.tb* infection

To further investigate local TAD DoD in detail, we explored the genome sites with dynamic DoD, such as the TAD with the guanylate-binding protein (GBP) gene family. The DoD value of this TAD was significantly decreased upon *M.tb* infection in the bulk cell DLO Hi-C data (Fig. 4a). The single-cell chromatin interaction matrix in this *GBP* region was merged as shown in Fig. 4b, in which each color represented the chromatin contact from the same individual cells. We calculated the DoD based on the merged chromatin interaction matrix from sciDLO Hi-C data and found that the DoD was decreased after *M.tb* infection, which is consistent with the results from bulk cell Hi-C data.

By virtue of the merged single-cell contact matrix, we found that the chromatin contacts fluctuate considerably among individual cells, suggesting heterogeneous genome organizational patterns at the single-cell level. However, upon infection, this TAD has smaller DoD and more intrinsically organized local chromatin interaction patterns in comparison to the random pattern in the Thp1-macro cells with high DoD (Fig. 4b). For example, a series of genome loci in this TAD were sequentially interacted with *GBP5* respectively among individual Thp1-*M.tb* cells, whereas the chromatin contact pattern of this location in Thp1-macro cells was much more random (Fig. 4c). As one Hi-C experiment can only capture chromatin interaction at single time-point, the merged interactions of *GBP5* with other loci from the individual cells may reflect the chromatin interactions around *GBP5* from different time-points. As they all looped with *GBP5*, it is likely that these genes are all assembled in a transcription factory.

Consistent with our observation that DoD is negatively correlated with gene co-regulation, Fig. 4a demonstrated that the expression levels of the *GBP1-5* in the TAD were synchronously co-upregulated. The immunostaining assay showed that GBP family proteins assembled tightly around the surface of *M.tb* cells probably to prevent its spreading (Supplementary Fig. 5a). These observations may suggest that this relatively chaotic TAD with high DoD or entropy before the infection became highly organized in response to *M.tb* infection. This may be coordinated by the opening of specific genome loci and the subsequent binding of corresponding proteins, which form new chromatin loops that link the genes into an active transcription factory. In this way, it could efficiently achieve synchronous co-transcription of these defense-related genes.

The transcription factory tends to form liquid–liquid phase separation (LLPS) condensates[43] to efficiently activate gene transcription. It has been shown that super-enhancer-binding proteins MED1 and BRD4 undergo phase separation and can be used as phase separation marker proteins within the transcription factory[44]. We then test whether *GBP* family gene region is relocated into LLPS zone in the process of immune activation. Based on previous BRD4 and MED1

ChIP-seq data[45], we found that *GBP* family genome regions were enriched with these two phase separation marker proteins (Fig. 4d). Our ATAC-seq and H3K4me3 ChIP-seq data indicate that the chromatin state around this region is activated and may bind with more regulatory proteins after infection (Fig. 4a). Furthermore, the integrated Hi-C chromatin loop and BRD4 and MED1 ChIP-seq analysis suggest that the BRD4 and MED1 occupied super-enhancers are spatially in close proximity to all the *GBP* family gene loci after *M.tb* infection (Fig. 4d).

To further confirm phase separation in *GBP* gene family region, we performed co-staining of BRD4 and MED1 (by Immunofluorescence) with *GBP* gene family region (by fluorescence in situ hybridization). As shown in Fig. 4e and Supplementary Fig. 5b, the overlapping ratio of BRD4 and MED1 puncta and the DNA-FISH signal of *GBP* gene family region is significantly increased after infection, suggesting the *GBPs* gene region tends to relocate into LLPS transcription factory zone to efficiently initiate this immune-defense gene expression. Together, our data implied that macrophages could dynamically adapt their 3D genome structure and coordinate with LLPS transcription factory to efficiently express this immune-defense gene to fulfill distinct physiological functions during immunological response.

## NF-kB initiates systematic chromatin remodeling of its target genome regions during *M.tb* infection

During differentiation and *M.tb* infection, the overall DoD was gradually decreased, which is in line with the plasticity of these cells (Fig. 5a and Supplementary Fig. 6a). In these processes, the reduction of DoD was accompanied with the increase of chromatin loops, suggesting that more loop-mediated *cis*-element interaction occurs in low DoD region (Fig. 5b). Upon infection, about 1864 loops were strengthened. These chromatin loops related genes are listed in Supplementary Data 4. Notably, these genes are significantly enriched in immunity pathways, which is not observed in the genes located in the weakened loops (Supplementary Fig. 6b). The dynamic immunity-related enhancer-gene regulatory network in Chr12 during infection was shown in Supplementary Fig. 6c. The KEGG pathway analysis of strengthened loop anchor genes (Supplementary Data 4) showed that 6 of the top 10 enriched pathways were directly related to NF-κB signaling pathways (Supplementary Fig. 6d). Furthermore, a large number of NF-κB binding motifs were enriched around the transcription start site (TSS) of these strengthened loop anchor genes (Fig. 5c), suggesting that NF-κB participates in chromatin remodeling during *M.tb* infection. The immunofluorescence assay revealed that NF-κB (p65) was indeed translocated into the nucleus upon infection (Fig. 5d).

The chromatin accessibility and RNA expression analysis showed that, upon *M.tb* infection, the NF-κB target loci turned to be more open, and the expression level of target genes was significantly upregulated compared to the random genes (Fig. 5e). Moreover, the chromatin loops related to the NF-κB target genes were more strengthened compared to the random genes (Fig. 5f). We further

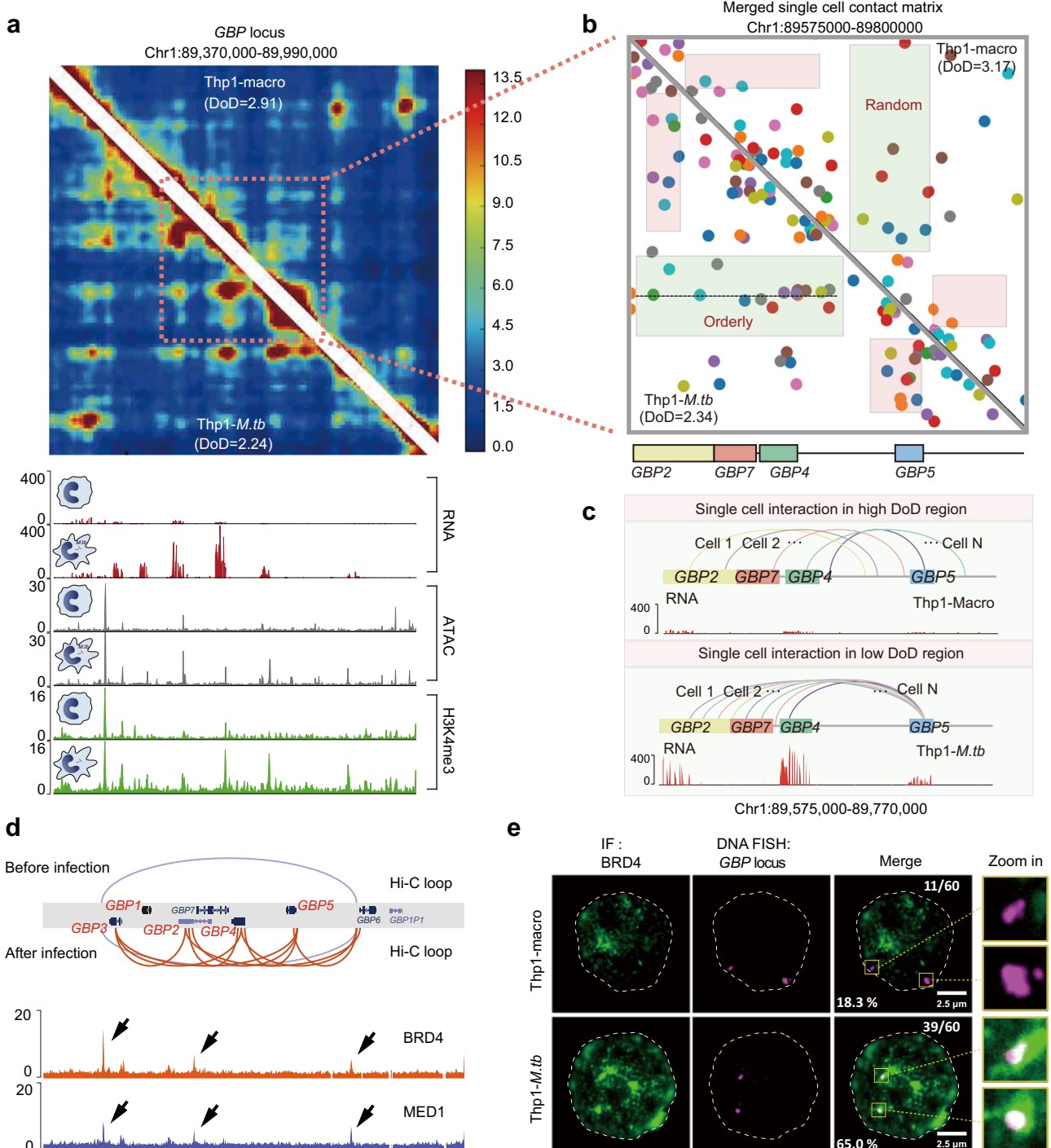

**Fig. 4 | Remodeling of the chromatin configuration of guanylate-binding protein (GBP) gene clusters.** **a** Comparison of TAD DoD value, chromatin interaction matrices, chromatin loops, RNA expression levels, and epigenetic modifications around the *GBP* loci in bulk cells before and after *M.tb* infection. **b** Comparing the single-cell chromatin contact differences using merged single-cell chromatin contact matrix. Each color of the dots in the matrix represents the chromatin contact from the same cell. **c** Single-cell chromatin contacts between *GBP2, GBP4, GBP5,* and *GBP7* and RNA expression before and after *M.tb* infection. **d** Depiction Hi-C chromatin loops before and after infection and BRD4 and MED1 ChIP-seq peaks around *GBP* gene clusters. **e** Colocalization between BRD4 and the *GBP* gene cluster by IF and DNA-FISH in Thp1-macro and Thp1-*M.tb* cells. IF, DNA-FISH, and merged channels (overlapping signal in white) are shown in separate images. The dashed line highlights the nuclear periphery, determined by DAPI staining. The rightmost column shows the area in the yellow box in greater detail. For each cell type, we counted 60 *GBP* gene loci. 18.3% (11/60) *GBP* loci were colocated with BRD4 in Thp1-macro cells and 65.0% (39/60) *GBP* loci were colocated with BRD4 in Thp1-*M.tb* cells. Each experiment was replicated three times.

investigated the chromatin remodeling of the typical NF-κB target gene loci, such as *IFITs, CCLs, GBPs, HERCs, NFKB1,* and *TNFSF10*. Figure 5g demonstrated that upon infection, a greater number of loops were formed in these regions, and the corresponding DoD was also reduced. This data was further validated by ChIP-qPCR, showing that the NF-κB was significantly enriched in the loop anchor regions of *IFIT3, CCL2, GBP4,* and *HERC2* upon infection (Supplementary Fig. 6e). This evidence suggests that, upon infection, NF-κB translocated into

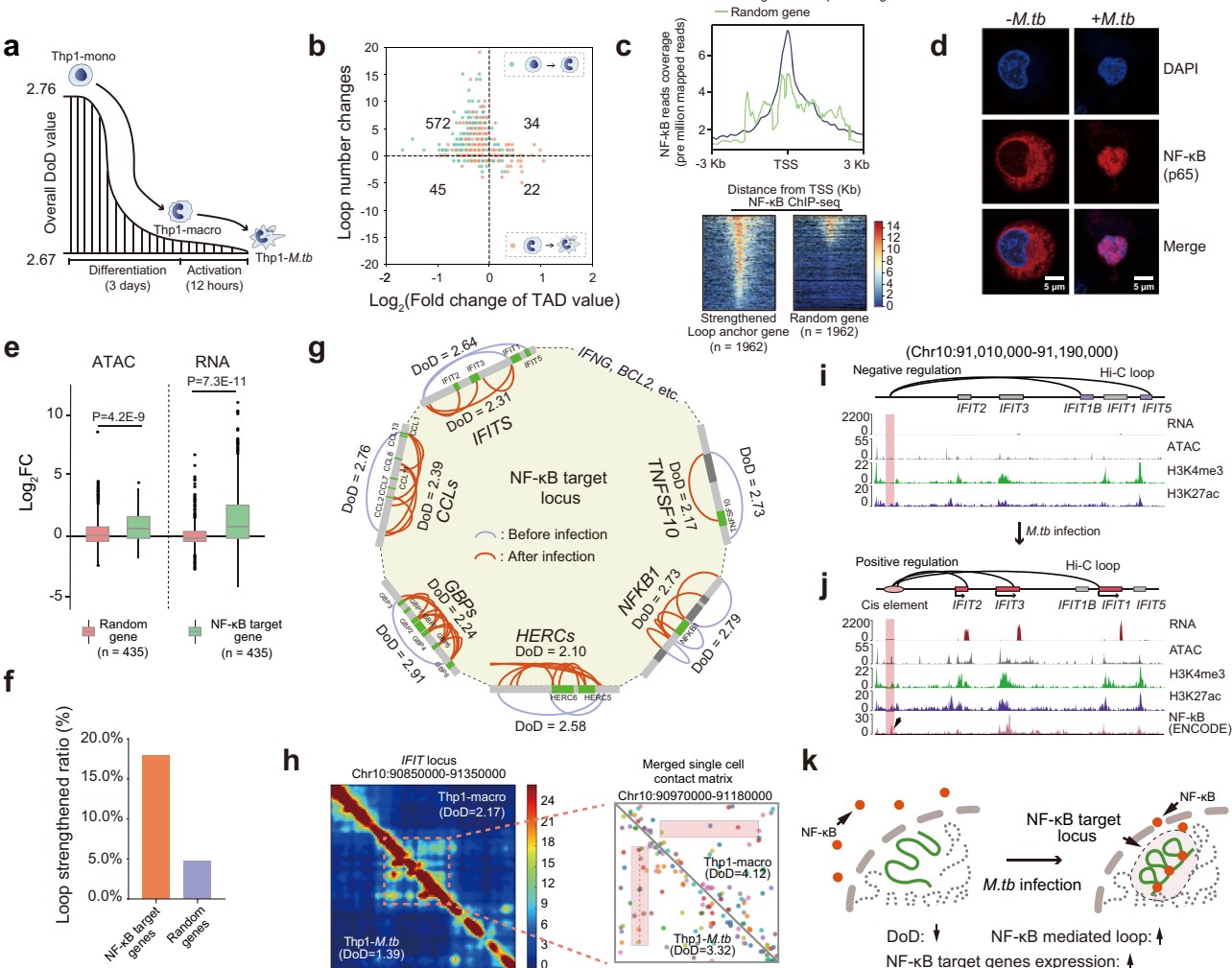

**Fig. 5 | Reorganization of chromatin architecture around NF-κB target gene sites. a** Overall TAD DoD of THP-1 cells during differentiation and infection. The *y*-axis represents the average DoD value of TADs in each state. The data are from Supplementary Fig. 3g. **b** The relationship between TAD DoD and chromatin loop during differentiation and activation. The *x*-axis represents log2 fold change of TAD DoD value, and the *y*-axis represent the number of loop changes in each TAD. **c** Compare NF-κB enrichment around transcription start site (±3 kb) between strengthened loop anchor gene and random control gene. The gene in the heatmap were sorted according to the enrichment intensity of NF-κB. ChIP-seq data are from ENCODE (GM12891 cell line). **d** Immunofluorescence analysis of NF-κB (p65) sub-cellular location before and after *M.tb* infection. Each experiment was replicated three times. **e** Log2 fold changes of ATAC peaks and RNA expression level of NF-κB target genes and random control genes. Box-plot with midline = median, box

limits = Q1 (25th percentile)/Q3 (75th percentile), whiskers = minimum and maximum values, points = outliers (>1.5 interquartile range). The sample sizes (n) are labeled in the figure. *P*-values were calculated by unpaired one-sided *t*-test. **f** The ratio of strengthened loop/total loop of NF-κB target gene sties and random control sites. **g** Chromatin loop remodeling and TAD DoD changes in typical NF-κB target gene loci. **h** Bulk cell and merged single-cell chromatin interaction matrices around *IFIT* gene locus. Cells with more consistent chromatin interaction were marked with a dashed line. **i, j** Comparison of chromatin loop configurations, RNA expression levels, and epigenetic modifications of the *IFIT* gene family locus before (**i**) and after (**j**) *M.tb* infection. The arrows in **j** indicate the potential *cis*-element linked with *IFIT2*, *IFIT3*, and *IFIT1*. **k** Schematic diagram of chromatin remodeling of NF-κB target genes during infection.

---

the nucleus, bound to specific target regions, readjusted the local DoD by reorganizing the chromatin structure for a concerted transcription of the defense genes.

*IFIT* gene loci are NF-κB target gene sites[46], of which the DoD was decreased upon infection. The single-cell Hi-C data also demonstrated that the local chromatin interaction pattern became more ordered during infection (Fig. 5h). The chromatin loop analysis revealed that these *IFIT* genes were regulated by a single upstream *cis*-element enriched with NF-κB binding motif (Fig. 5i, j). Before infection, the *cis*-element was linked to *IFIT1B* and *IFIT5* (Fig. 5i), which were dynamically reshuffled to link with *IFIT1*, *IFIT2*, *IFIT3* after infection (Fig. 5j). Interestingly, it has been demonstrated that IFIT1, IFIT2, and IFIT3 proteins can interact with each other and form a complex to perform antipathogenic functions[47]. Consistently, we observed synchronously

upregulated expression of these three genes in the newly established loops (Fig. 5j). Notably, *IFIT1B* and *IFIT5*, which are not located in this spatially adjacent hub, were not upregulated. These data demonstrated a high-order chromatin structure mediated elegant spatial and temporal co-regulation of the NF-κB target genes transcription during immunoresponse (Fig. 5k).

Next, we collected monocytes from peripheral blood and induced them into hMDMs by macrophage colony-stimulating factor (M-CSF). After virulent tuberculosis strain H37Rv infection, the RNA-Seq and DLO Hi-C libraries of these hMDMs-*M.tb* cells were constructed. As shown in Supplementary Fig. 7a, the differentially expressed genes in Thp1-*M.tb*, such as *GBP1*, *GBP5*, *IFIT3*, *CCR7*, and *PD-L1* (Supplementary Data 2), were also significantly upregulated in hMDMs-*M.tb*. The majority of the enriched KEGG pathways, such as NF-κB signaling

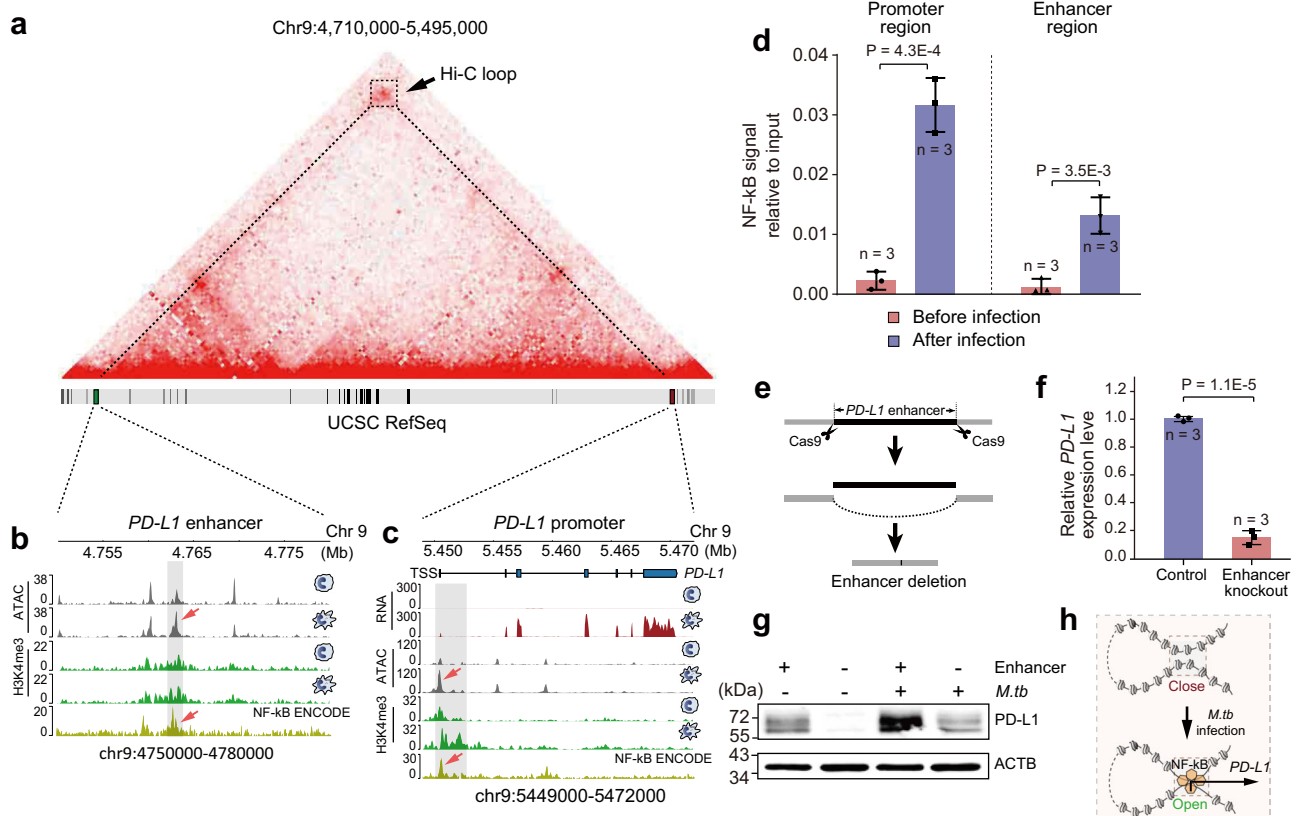

**Fig. 6 | Functional identification of *PD-L1* enhancer. a** Interaction of the *PD-L1* enhancer and promoter in the Hi-C matrix. **b, c** Genome Browser view of RNA expression levels, chromatin accessibility, and histone modifications at the PD-L1 gene and enhancer regions in Thp1-macro and Thp1-*M.tb* cells. NF-κB ChIP-seq peaks (ENCODE, GM15510 cell line) and strengthened ATAC-seq peaks were marked by arrows. **d** ChIP-qPCR validation of the NF-κB (P65) enrichment on the *PD-L1* enhancer and promoter regions. The amount of immunoprecipitated DNA in each sample is represented as signal relative to the total amount of input chromatin (*y*-axis). Error bars show mean ± SD (standard deviation), *n* = 3 biologically

independent samples. *P*-values were calculated by two-sided Student's *t*-test. **e** Experimental design of sgRNA-guided enhancer perturbation by the Cas9 protein. **f** Relative mRNA expression levels of *PD-L1* in Thp1-macro cells and the same line after enhancer deletion. Error bars show mean ± SD, *n* = 3 biologically independent samples. *P*-values were calculated by two-sided Student's *t*-test. **g** PD-L1 protein levels in Thp1-macro cells and the same cell line after enhancer deletion before and after *M.tb* infection. Each experiment was replicated three times. **h** Schematic of enhancer-promoter interaction mediated regulation of *PD-L1* expression during *M.tb* infection.

pathway, TNF signaling pathway, and Jak-STAT signaling pathway, are consistent between hMDMs-*M.tb* and Thp1-*M.tb* (Supplementary Fig. 7b, c).

Furthermore, we validated the TAD DoD dynamics of NF-κB target loci after *M.tb* infection in hMDM cells, the TAD DoD value of hMDMs-*M.tb* decreased significantly (Supplementary Fig. 7d, *P* = 3.4E-13) compared to the control group, especially in *GBP2*, *GBP5*, *NFKB1*, and *TNFSF15* gene loci (Supplementary Fig. 7e). Consistent with THP-1-derived macrophages (Fig. 4a and Fig. 5g), the DoD value of *GBP* gene locus decreased from 2.98 to 1.83 after *M.tb* infection in hMDMs (Supplementary Fig. 7f, g). These results suggest that the immunoresponse of THP-1 infection model is highly similar to primary human macrophages.

### A remote NF-κB enriched enhancer promotes the expression of *PD-L1* through a chromatin loop

After engulfing *M.tb*, macrophages can present *M.tb* antigens via major histocompatibility complex (MHC) molecules to T-cells to eliminate *M.tb* infection[48]. During this process, *M.tb* also evolves various strategies to escape immunological clearance. In this scenario, we observed the expression of immune checkpoint gene *PD-L1* was dramatically upregulated after *M.tb* infection (Fig. 1c). Through the Hi-C contact matrix, we identified a putative enhancer region highly enriched with H3K4me3 modification (Chr9: 4,760,070–4,769,779) contact with the *PD-L1* promoter through a Hi-C loop (Fig. 6a–c). Notably, the ATAC-seq

data showed that these enhancer and promoter regions were more accessible, and enrichment of active chromatin modifications after *M.tb* infection (Fig. 6b, c), indicates that more regulatory proteins are enriched in the enhancer region. Transcription factor binding motif analysis revealed that both enhancer and promoter regions (Fig. 6b, c and Supplementary Fig. 8a, b) can be bound by NF-κB. Furthermore, by ChIP-qPCR, we proved that NF-κB (p65) was significantly enriched in this enhancer region after infection (Fig. 6d).

Since NF-κB can directly bind to *PD-L1* promoter and upregulate its gene transcription[49], we speculate that this enhancer region can directly regulate the *PD-L1* gene expression via chromatin loop. To further confirm the function of *PD-L1* enhancer, we knocked out this enhancer region in the THP-1 cell line by the CRISPR/Cas9 system and validated its regulatory function (Fig. 6e and Supplementary Fig. 8c–f). As predicted, the qPCR and western blot data demonstrated that knockout of the *PD-L1* enhancer can indeed attenuate the upregulation of *PD-L1* expression at both mRNA and protein levels upon *M.tb* infection (Fig. 6f, g). Collectively, these results suggested a synchronous opening of the enhancer and promoter regions for transcription factor binding and activated *PD-L1* gene transcription during *M.tb* infection (Fig. 6h). This *PD-L1* enhancer has the potential to become anti-tuberculosis and even antitumor therapeutic target. It would be of great interest to further investigate how *M.tb* infection reprograms the epigenetic code in this enhancer locus and what is the physiological role of this modification in the pathogenesis of *M.tb*.

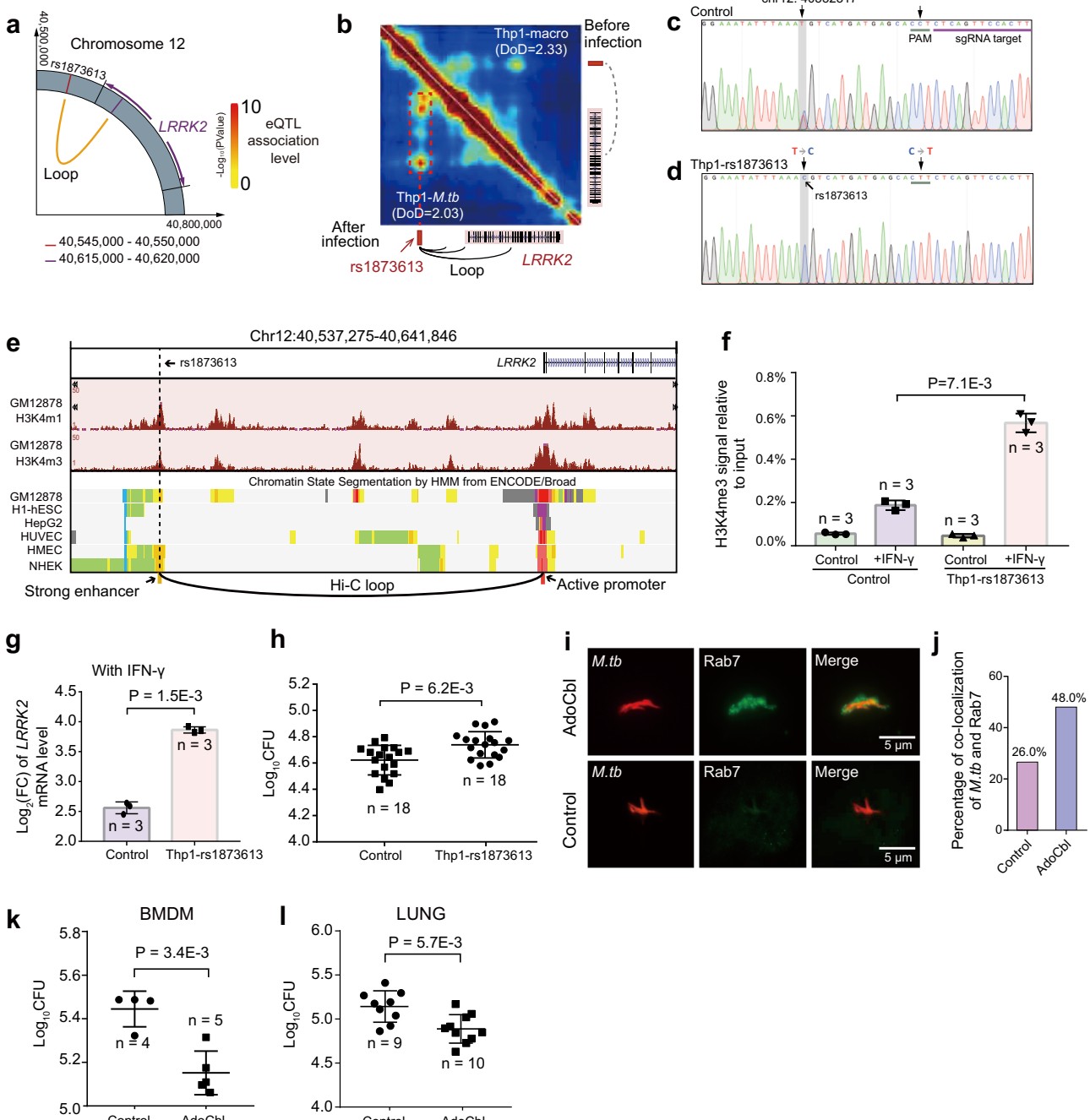

**Fig. 7 | Identification of long-range regulatory target genes of mycobacterial disease susceptibility loci. a** Identification of long-range regulatory target gene of rs1873613. **b** The rs1873613 is located in the anchor of the dynamic *LRRK2* chromatin loop. **c** Sanger sequencing result of the chr12: 40552317 loci in THP-1 control cell line. **d** Validation of "T" to "C" single base pair mutation in chr12:40552317 (rs1873613). **e** Histone modification, chromatin state, and Hi-C loop information of rs1873613 and *LRRK2* gene region from the UCSC Genome browser (http://genome. ucsc.edu). **f** ChIP-qPCR validation of the H3K4me3 enrichment in the *LRRK2* enhancer region. The amount of immunoprecipitated DNA is represented as signal relative to the total amount of input chromatin (*y*-axis). Error bars show mean ± SD, *n* = 3 biologically independent repeats. *P*-values were calculated by two-sided Student's t-test. The final concentration of IFN-γ is 20 ng/ml. **g** Log₂ fold change of *LRRK2* mRNA expression in THP-1 control and the Thp1-rs1873613 cell line under IFN-γ stimulation (20 ng/ml). Error bars show mean ± SD, *n* = 3 biologically

independent repeats. *P*-values were calculated by two-sided Student's t-test. **h** CFU assays of *M.tb* in THP-1 control and Thp1-rs1873613 cell lines under IFN-γ stimulation (20 ng/ml). Error bars show mean ± SD, *n* = 18 biologically independent samples were used for the control group and Thp1-rs1873613 group. *P*-values were calculated by unpaired one-sided t-test. **i, j** Immunofluorescence analysis and quantification of the colocalization of the *M.tb* (H37Ra-RFP) and Rab7 in BMDM treated with the LRRK2 inhibitor AdoCbl and PBS (control), respectively. **k** CFU assays of *M.tb* in BMDM treated with AdoCbl and PBS (control), respectively. Error bars show mean ± SD, *n* = 4 and *n* = 5 biologically independent samples were used for the control group and AdoCbl treatment group, respectively. *P*-values were calculated by unpaired one-sided t-test. **l** CFU assays of *M.tb* in C57BL/6 mouse lungs. Error bars show mean ± SD, *n* = 9 and *n* = 10 mice were used for the control group and AdoCbl treatment group, respectively. *P*-values were calculated by unpaired one-sided t-test.

### Identification of long-range target genes of mycobacterial disease susceptibility loci via chromatin loop

To obtain a comprehensive map of mycobacterial disease susceptibility loci and their long-range regulatory gene targets via chromatin loop, we collected all reported susceptibility loci[21–28] and performed integrated omics analysis of GWAS, eQTL, and Hi-C (Fig. 7a and Supplementary Fig. 9a and Supplementary Data 5). With DLO Hi-C data, we could systematically identify the target genes of these susceptibility loci through long-range chromatin interactions (Supplementary Fig. 9b). Of note, the chromatin interaction loops between the *LRRK2, NSL1*, and *ASAP1* and the corresponding susceptibility loci were significantly strengthened during *M.tb* infection (Fig. 7b and Supplementary Fig. 9c, d). Importantly, the target genes of susceptibility loci discovered by this integrated omics analysis, such as *ASAP1* and *LRRK2*, have been reported to be involved in mycobacterial pathogenesis[23,50], supporting the integrity of our multi-omics analysis. Further, it would be of great importance to obtain GWAS raw data and perform a colocalization analysis of the GWAS and the eQTL signals.

We further analyzed *LRRK*, which has a loop and eQTL correlation with the susceptibility SNP rs1873613. The DoD value of *LRRK2* locus was decreased after *M.tb* infection (Fig. 7b). Moreover, the simulated chromatin structure based on single-cell Hi-C data revealed that rs1873613 and *LRRK2* tended to be adjacent in space upon infection (Supplementary Fig. 10a). Furthermore, this SNP is located right in the anchor of an *LRRK2* chromatin loop that was significantly strengthened after infection (Fig. 7b). This data suggests that this susceptibility SNP locus was dynamically reshuffled upon infection to spatially link with *LRRK2* to regulate its gene transcription.

To further explore the function of SNP rs1873613, we introduced a "T" to "C" single base pair mutation in chr12:40552317 using the CRISPR/Cas9 system in THP-1 cell lines and named the mutated cell line as Thp1-rs1873613 (Fig. 7c, d). Through the ENCODE ChromHMM chromatin state information from UCSC browser, we found that rs1873613 is located in a strong enhancer upstream of *LRRK2*. This enhancer interacts with *LRRK2* promoter through a long-range chromatin loop (Fig. 7e). Therefore, we speculated that rs1873613 would affect the activity of the enhancer, thereby regulating the expression of *LRRK2* gene.

Since *LRRK2* is an IFN-γ target gene, we stimulated THP-1 control and Thp1-rs1873613 with IFN-γ, respectively, and then evaluated the enhancer activity by ChIP-qPCR. As shown in Fig. 7f, under IFN-γ stimulation, the Thp1-rs1873613 had significantly more ($P = 0.0071$) active chromatin mark (H3K4me3) enriched in the enhancer region compared to the THP-1 control. This data indicates that the enhancer region of Thp1-rs1873613 is more sensitive to IFN-γ and has stronger enhancer activity than that of the THP-1 control. Moreover, rs1873613 significantly promoted the inductive effect of IFN-γ on *LRRK2* expression (Fig. 7g).

Consistently, the eQTL analysis data also showed that a T:C mutation can indeed increase the expression of *LRRK2* in the lungs (Supplementary Fig. 10b). Since the expression of *LRRK2* is positively correlated with the intracellular survival of *M.tb*[50], we therefore investigated the intracellular proliferation of *M.tb* in THP-1 control and Thp1-rs1873613 cell lines. As shown in Fig. 7h, the bacterial load of Thp1-rs1873613 was indeed significantly higher than that of the THP-1 control group. Taken together, we verified that rs1873613 can enhance the *LRRK2* enhancer activity and upregulate the expression of the *LRRK2*, further supporting the correlation of this SNP to mycobacterial disease susceptibility. In the future, it would be of great importance to perform more in vivo experiments to further evaluate the causal mechanisms of rs1873613 and other SNPs in mycobacterial disease susceptibility.

It has been demonstrated that LRRK2 promoted the proliferation of *M.tb* by inhibiting phagosome maturation[50], suggesting LRRK2 might be a TB drug target. Thus, we investigated the anti-*M.tb* effect of AdoCbl, an inhibitor of LRRK2[51]. As shown in Fig. 7i–k, AdoCbl can indeed promote the maturation of phagosomes in TB-infected bone-marrow-derived macrophages (BMDM) and decrease the count of colony-forming units of intracellular *M.tb*. Furthermore, the in vivo experiment showed that AdoCbl could significantly inhibit the proliferation of *M.tb* in the lungs (Fig. 7l and Supplementary Fig. 10c). This result was further supported by hematoxylin-eosin staining of tissue sections, showing that AdoCbl significantly inhibited the initiation of lung and spleen lesions (Supplementary Fig. 10d). These data revealed that the TB susceptibility SNP rs1873613 in the dynamic loop anchor of *LRRK2* enhancer could promote the *LRRK2* enhancer activity, thereby upregulating the expression of the *LRRK2*. LRRK2 protein subsequently represses the clearance of *M.tb* by inhibiting phagosome maturation. We demonstrated that this process can be attenuated by a potential TB drug candidate, AdoCbl (Supplementary Fig. 10e).

Collectively, we delineated the 3D genome landscape of THP-1 cells during differentiation and infection at single-cell resolution. Our data showed that the immunological enhancer-promoter loops, especially the NF-kB target regions, were reorganized to orchestrate synchronous defense gene transcription for *M.tb* clearance. It provides a comprehensive resource for epigenetic regulation of immunocytes and for *M.tb* infection studies, such as anti-*M.tb* drug screening and the TB pathogenesis mechanisms of the patients with susceptibility SNPs. Importantly, we proposed TAD DoD to measure the genome organizational patterns, which are correlated with the chromatin epigenetic states, chromatin structure variability in individual cells, and expression and co-regulation of the genes within the TAD, supporting that the order and stochasticity of genome architecture are related to its function. These data shed insights into the dynamic genome organizational patterns at single-cell level and illustrated how the spatial organization of chromatin coordinates gene transcriptional programs during differentiation or in response to different stimulus, such as immunological response.

## Methods

### Cell culture

THP-1 cells (ATCC, TIB-202) were grown in RPMI-1640 containing 10% FBS. To differentiate the monocytes from macrophages, cells were treated with PMA (phorbol 12-myristate 13-acetate, sigma, Catalog code: P8139) for 48 h at a final concentration 40 ng/ml, and then washed with prewarmed PBS and incubated with fresh culture medium for another 24 h.

Human peripheral blood total monocyte was isolated using MagniSort™ Human pan-Monocyte Enrichment Kit (Invitrogen, Catalog code: 8804-6837-74). To differentiate the human pan-monocyte to human monocyte-derived macrophages (hMDMs), cells were treated with human M-CSF (macrophage colony-stimulating factor, Chamot Biotechnology, Catalog code: CM116-20MP) for 10 days at a final concentration 50 ng/ml, and then washed with prewarmed PBS and incubated with fresh culture medium for another 24 h.

### Mycobacterium tuberculosis infection

For *M.tb* infection experiment, H37Ra or H37Rv was pelleted ($3000 \times g$, RT, 10 min), washed twice with RPMI-1640, resuspended in 1 ml THP-1 culture medium, and dispersed using BD insulin syringes (BD, Catalog code: 328421). THP-1-derived macrophages (Thp1-macro) or human monocyte-derived macrophages (hMDMs) were infected with H37Ra or H37Rv at MOI (multiplicity of infection) 20. Four hours later the infected cells were washed twice with prewarmed PBS and incubated with a fresh culture medium for another 8 h.

### RNA-seq library preparation

RNA was extracted using the RNAiso Plus (Takara, Catalog code: 9109) according to the manufacturer's protocol. Sequencing libraries were

prepared using the VAHTS Stranded mRNA-Seq Library Prep Kit (Vazyme, Catalog code: NR602-02) according to the manufacturer's protocol.

## ChIP-Seq library preparation

$4 \times 10^6$ cells were used per immunoprecipitation. Antibody of H3K4me1 (Abcam, Catalog code: ab8895, 1:100 dilution), H3K4me3 (Abcam, Catalog code: ab8580, 1:100 dilution), H3K9me3 (Abcam, Catalog code: ab8898, 1:100 dilution), H3K27ac (Abcam, Catalog code: ab4729, 1:100 dilution), H3K27me3 (Millipore, 07-449, 1:100 dilution), and NF-κB p65 (CST, Catalog code: 8242, 1:100 dilution) were used in this study. All antibody dilutions were 1:100. The ChIP DNA was obtained using SimpleChIP® Enzymatic Chromatin IP Kit (CST, Catalog code: #9003). After immunoprecipitation, ChIP-Seq library was constructed using NEBNext® Ultra™ DNA Library Prep Kit (NEB, Catalog code: E7370S). In addition, the ChIP DNA was also used as the template for ChIP-qPCR.

## ATAC-seq library preparation

$1 \times 10^5$ cells were centrifuged at $800 \times g$ for 5 min and then washed once using 500 µl of cold 1× PBS and centrifuged at $800 \times g$ for another 5 min. Cells were lysed using cold lysis buffer (10 mM Tris-HCl (pH 8.0 at 25 °C), 10 mM NaCl, 0.3% Igepal CA-630). After lysis, nuclei were spun down at $800 \times g$ in 4 °C for 10 min. The pellet was resuspended in the transposase reaction mix (10 µl 5× TTBL buffer (Vazyme, Catalog code: TD501-02), 5 µl TTE Mix V50 buffer (Vazyme, Catalog code: TD501-02), and 35 µl ddH₂O water). Tagmentation was carried out for 10 min at 55 °C. Immediately following transposition, the DNA was purified using a Qiagen MinElute PCR Purification Kit (Qiagen, Catalog code: 28004) and eluted with 10 µl elution buffer. The ATAC-Seq library was amplified by the primers in TruePrepTM Index Kit V2 for Illumina® (Vazyme, Catalog code: TD202).

## In situ DLO Hi-C experiment

Five million cells were double crosslinked with 1.5 mM EGS (Sigma, Catalog code: E3257) and 1% formaldehyde (Sigma, Catalog code: F8775) and lysed in lysis buffer (10 mM Tris-HCl (pH 8.0 at 25 °C), 10 mM NaCl, 0.3% Igepal CA-630, 0.5% SDS, and complete protease inhibitor (Roche)), incubated at 60 °C for 5 min, and placed on ice immediately. After incubation, the nuclei were pelleted by centrifugation at $1800 \times g$. for 5 min and washed once with ice-cold PBS. A total of 310 µl of ddH₂O, 20 µl of 20% Triton X-100, 40 µl of 10× NEBuffer 2.1, and 30 µl of MseI (NEB, Catalog code: R0525L, 10 units/µl) was then added to the nuclei and incubated for 6 h at 37 °C with rotation at 15 r.p.m. After restriction enzyme digestion, 50 µl of MseI half linkers (600 ng/µl), 5 µl of 100 mM ATP, 20 µl of T4 DNA ligase (Thermo, Catalog code: EL0011, 5 units/µl), and 25 µl of ddH₂O were added to the 400 µl of digested chromatin and mixed thoroughly. The mixture was then incubated at 25 °C for 1 h with rotation at 15 r.p.m. After half-linker ligation, the nuclei were centrifuged at 4 °C for 5 min at 1000 r.p.m. and washed twice with 1 ml of ice-cold PBS. The linker-ligated nuclei were gently resuspended in 200 µl of 1× T4 DNA ligation buffer (Thermo) containing 0.5 units/µl T4 polynucleotide kinase (NEB) and incubated at 37 °C for 30 min. The 200 µl reaction complexes were added to 300 µl of 1× T4 DNA ligation buffer (Thermo) containing 0.5 units/µl T4 DNA ligase (Thermo, Catalog code: EL0011). Ligation was performed at 20 °C for 2 h with rotation at 15 r.p.m. The nuclei were centrifuged at 4 °C for 5 min at $1800 \times g$ and resuspended in 400 µl of ddH₂O. Protein digestion was performed by adding 25 µl of 10 mg/ml proteinase K (Sigma, Catalog code: 39450-01-6), 50 µl of 10% SDS, and 25 µl of 5 M NaCl, and the tubes were incubated for 2 h at 65 °C. After incubation, an equal volume of phenol:chloroform:isoamyl alcohol (25:24:1) was added to the sample, shaken vigorously, and then centrifuged for 10 min at $14,000 \times g$. Next, the supernatant was transferred to a new tube. This process was repeated twice. DNA was

precipitated at room temperature with 5 µl of Dr. GenTLE Precipitation Carrier (Takara, Catalog code: 9094), 50 µl of 3 M sodium acetate (pH 5.2), and 555 µl of isopropanol. The precipitated DNA was washed once with 80% ethanol and dissolved in 160 µl of ddH₂O. A total of 20 µl of 10× CutSmart buffer, 10 µl of SAM (NEB, Catalog code: B9003S), and 10 µl of MmeI (NEB, Catalog code: R0637L, 2 units/µl) were added to the 160 µl DNA sample, and digestion was performed at 37 °C for 1 h. The digested DNA sample was subjected to electrophoresis in native PAGE gels. The specific 80 bp DLO Hi-C DNA fragments were excised and transferred to a 0.6 ml tube with a pierced bottom. This tube was then placed into a 1.5 ml tube, and the gel slices were shredded by centrifugation at $14,000 \times g$. for 10 min. A total of 400 µl of TE buffer was added to the 1.5 ml tube (to ensure that the shredded gel was fully immersed in buffer), and the mixture was incubated for 20 min at −80 °C, followed by a 2 h incubation at 37 °C with rotation at 15 r.p.m. Next, the shredded gel, along with the buffer, was transferred into the filter cup of a 2 ml Spin-X tube filter (Costar, Catalog code: 8160). After centrifugation, the eluate was transferred into a new 2 ml tube. A total of 4 µl of Dr. GenTLE Precipitation Carrier (Takara, Catalog code: 9094), 40 µl of 3 M sodium acetate (pH 5.2), and an equal volume of isopropanol were added to precipitate the DNA. The precipitated DNA was washed once with 80% ethanol and dissolved in 40 µl of ddH₂O. Next, 1.5 µl of PE-adaptor1 (500 ng/µl), 1.5 µl of PE-adaptor2 (500 ng/µl), 5 µl of 10× T4 DNA ligase buffer, and 2 µl of T4 DNA ligase (Thermo, Catalog code: EL0012) were added to the 40 µl of DLO Hi-C DNA fragments and incubated at 16 °C for approximately 30 min. 90 µl of AMPure XP beads (Beckman, Catalog code: A63880) was added to the ligation mixes and washed twice with 80% ethanol to remove excess Illumina sequencing adaptors. Next, 45 µl of ddH₂O was used to wash DNA from the beads. The eluted DNA was repaired using PreCR Repair Mix (NEB, Catalog code: M0309S) for 20 min at 37 °C in a final volume of 50 µl. 5-10 µl of repaired DNA was used as a template and amplified for fewer than 13 cycles. The PCR product is the final DLO Hi-C sequencing library.

## Single-cell DLO Hi-C experiment

**Nuclei preparation.** Five million cells were crosslinked with 1% formaldehyde (Sigma, Catalog code: F8775) for 10 mins, and lysed in lysis buffer (10 mM Tris-HCl (pH 8.0 at 25 °C), 10 mM NaCl, 0.3% Igepal CA-630, 0.5% SDS, and complete protease inhibitor (Roche)) at 60 °C for 5 min, and placed on ice immediately. After incubation, the nuclei were pelleted by centrifugation at $1800 \times g$ for 5 min and washed once with nuclei wash buffer (PBS, which contain 0.5% Triton X-100, 0.05% Tween 20, and 0.05% CA-630).

**MseI digestion.** The digestion buffer (a total of 310 µl of ddH₂O, 20 µl of 20% Triton X-100, 40 µl of 10× NEBuffer 2.1, and 30 µl of MseI (NEB, Catalog code: R0525L, 10 units/µl) was then added to the nuclei pellet and digested for 3 h at 37 °C in thermomixer (Eppendorf) with rotation at 1000 r.p.m.

**Indexed half-linker ligation.** After digestion, 560 µl 2.1× T4 DNA ligase buffer were added to the nuclei, divided into 96-well plates, and adjusted to the volume of 10 µl/tube. Next, 1 µl barcoded half linkers (50 µM/µl) (Supplementary Data 6) were added to each tube and mixed well. After incubation at room temperature for 5 min, 1 µl T4 DNA ligase (Thermo, Catalog code: EL0011, 5 units/µl) was added to each tube. The ligation reaction was performed at 20 °C for 30 min, and then incubated for 10 min at 4 °C. The sample in 96-well plates was transferred to 1.5 ml DNA LoBind Tube (Eppendorf, Catalog code: B148089M), centrifuged at $1800 \times g$ for 5 min at 4 °C, and washed 4 times with nuclei wash buffer.

**Fragment-end phosphorylation and in situ proximity ligation.** The linker-ligated nuclei were gently resuspended in 200 µl of 1× T4 DNA

ligation buffer (Thermo) containing 0.5 units/µl T4 polynucleotide kinase (NEB, Catalog code: M0201L) and then incubated at 37 °C for 30 min. The 200 µl reaction complexes were added to 300 µl of 1× T4 DNA ligation buffer (Thermo) containing 0.5 units/µl T4 DNA ligase (Thermo, Catalog code: EL0011). Ligation was performed at 20 °C for 2 h with rotation at 15 r.p.m. The nuclei were centrifuged at 4 °C for 5 min at 1800 × $g$ and resuspended in nuclei wash buffer.

**Single-cell selection and multiple displacement amplification.** The nuclei were stained by DAPI (Thermo, Catalog code: D1306) and diluted by nuclei wash buffer. The density of the nucleus in liquid was carefully checked under the fluorescence microscope, and adjusted to 30 nuclei/µl. Next, about 1 µl nuclei were digested with proteinase K to release the DNA, the reaction system is as follows: 8 µl ddH$_2$O, 2 µl 10× Phi29 MAX DNA Polymerase Reaction Buffer (Vazyme, Catalog code: N106), 5 µl 5 N random primer (100 µM), 2 µl dNTP (10 mM each), 1 µl nuclei (~30 nuclei/µl), and 1 µl proteinase K (20 mg/ml). The reaction was performed at 60 °C for 1 h, 98 °C for 10 min, and 4 °C for 5 min. After proteinase K digestion, 1 µl Phi29 MAX DNA Polymerase (Vazyme, Catalog code: N106) was added to the sample, mixed well, and incubated at 30 °C for 3 h.

**Multiple displacement amplification recycle and sequencing library construction.** After multiple displacement amplification (MDA) reaction, 1 µl 10% SDS and 79 µl ddH$_2$O were added to the sample, and the DNA was purified by 200 µl VAHTS DNA Clean Beads (Vazyme, Catalog code: N411-01). The following steps of MmeI digestion, 80 bp contact DNA fragment recovery are the same as in situ DLO Hi-C protocol described above. For sequencing library construction, we replaced the previous Illumina sequencing adapter with a homemade MGI-2000 platform sequencing adapter with the second round of indexes (Supplementary Fig. 4c and Supplementary Data 6). Previous similar methods[52,53] have shown that due to the number of barcodes (96 × 14) being far more than the number of nuclei (30 × 14), therefore most of the single nuclei are labeled by a unique barcode.

**Immunofluorescence combined with DNA-FISH**
Cells grown on coated glass were fixed in 4% formaldehyde solution (Sigma, Catalog code: 47608-250ML-F) at room temperature for 5 min followed by PBS buffer washing for three times. Next, cells were permeabilized with 0.4% SDS in PBS for 5 min at RT. Following three washes in PBS for 5 min, cells were blocked in blocking buffer (1× PBS / 5% normal serum / 0.3% Triton X-100) for 60 min. After blocking, antibodies (anti-BRD4, Abcam, Catalog code: ab128874, 1:250 dilution; anti-MED1, Abcam, Catalog code: ab64965, 1:250 dilution) in antibody dilution buffer (1× PBS / 1% BSA / 0.3% Triton X-100) at 2 µg/ml final concentration were incubated with the slides overnight at 4 °C. Slides were then washed three times with PBS and recognized by secondary antibodies (Goat antiRabbit IgG Alexa Fluor 488, Life Technologies, Catalog code: A11008, 1:1000 dilution) in the dark for 30 min.

After immunofluorescence, cells were placed in prewarmed PBS and incubated at 60 °C for 20 min, then incubated in 70% ethanol, 85% ethanol, and then 100% ethanol for 1 min at RT. After alcohol dehydration, the cells were heated on a hot plate at 82 °C for 10 min in 80% formamide (Sigma, Catalog code: 47671-1L-F) and 2× SSC for DNA denaturation. Next, cells were incubated for 12 h in hybridization solution with 2 µM Alex555-dUTP labeled *GBP* DNA-FISH probes (Chr1: 89,448,890- 89,739,345, Spatial FISH Co. Ltd.) in the presence of 50% formamide, 8% dextran sulfate sodium salt (Sigma), and 2× SSC. After hybridization, the cells were washing for three times with 30% formamide and three times with 2× SSC. Next, the slides were stained with DAPI (Thermo, Catalog code: D1306) and observed under a super-resolution microscope (Nikon, N-SIM).

**CFU assays**
Bone-marrow-derived macrophage cells (BMDM) were seeded into six-well plate (1 × 10$^5$ / well) and infected with H37Ra at MOI (multiplicity of infection) 20. Four hours later, the infected cells were washed twice with prewarmed PBS and incubated with a fresh culture medium. Then the cells were treated with AdoCbl (Sigma, Catalog code: C0884) at a final concentration of 200 µM. The PBS solution was used as a negative control. Three days later, the cells were lysed with 0.1% triton X-100 and plated for bacterial burden enumeration using a serial dilution method on plates of Middlebrook 7H11 agar containing OADC enrichment (BD, Catalog code: 211886) and BBL MGIT PENTA antibiotics (BD, Catalog code: 245114). CFUs were counted after 3–4 weeks of incubation at 37 °C.

**Immunofluorescence**
For immunofluorescence of GBP1-5, Rab7, and NF-κB, monocyte-derived macrophages (Thp1-macro) were infected with RFP-H37Ra at MOI (multiplicity of infection) 20. The infected cells were washed twice with prewarmed PBS 4 h and incubated with a fresh culture medium for another 8 h. Next, the cells were crosslinked with 4% formaldehyde for 15 min at room temperature and rinsed three times with 1× PBS. The specimen was blocked in blocking buffer (1× PBS / 5% normal serum / 0.3% Triton X-100) for 60 min. After blocking, respective antibodies (GBP1-5, Santa Cruz, Catalog code: sc-166960 AF488, 1:250 dilution; Rab7, Santa Cruz, Catalog code: sc-376362 AF488, 1:250 dilution; NF-κb p65, Santa Cruz, Catalog code: sc-8008 AF546, 1:250 dilution) in antibody dilution buffer (1× PBS / 1% BSA / 0.3% Triton X-100) at 2 µg/ml final concentration were incubated with the slides overnight at 4 °C. Slides were washed three times with PBS and stained with DAPI (Life Technologies) and observed under the fluorescence microscope.

**Genome editing in THP-1 cells by CRISPR/Cas9 system**
In order to introduce the "T" to "C" mutation at the genome site chr12:40552317 (rs1873613) into THP-1 cells, we designed a sgRNA (sequence: "CTCAGTTCCACTTCTTACTC") to target chr12:40552317. Next, we constructed the homologous DNA fragment for homologous recombination. To avoid the cleavage of the homologous recombination arms of the DNA donor by Cas9 upon transfection, we also mutated the PAM sequence of the sgRNA target (chr12:40552317, "C" to "T") in the donor plasmid. In the homologous fragment, we simultaneously mutated the chr12: 40552317 site ("T" to "C") and PAM sequence of the sgRNA (chr12: 40552332, "C" to "T"). Then, we ligated the homologous DNA fragment, sgRNA cassette, GFP fluorescent protein gene, and puromycin-resistance gene into PUC19 plasmid. The structure of the PUC19-sgRNA-GFP-Puro-rs1873613 plasmid is illustrated in Supplementary Fig. 11.

Next, we transfected the PUC19-sgRNA-GFP-Puro-rs1873613 plasmid into Thp1-cas9 cell line (THP-1 cell line with stably expressing Cas9) using Nucleic Acid Transfection Enhancer (NATE, InvivoGen, Catalog code: lyec-nate) and PolyJet (SignaGen, Catalog code: SL100688). After 36 h of puromycin (final concentration: 2.5 µg/ml) selection, single cells were seeded into 96-well plates. During colony expansion, genotyping was carried by PCR and Sanger sequencing to screen single-cell clones which carried rs1873613 (Fig. 7c, d). We named the homozygote containing "T" to "C" mutation at chr12: 40552317 sites (rs1873613) as Thp1-rs1873613. Two days after IFN-γ stimulation (final concentration 20 ng/ml), the enrichment of H3K4me3 in *LRRK2* enhancer region and *LRRK2* mRNA expression levels were investigated by ChIP-qPCR and RT-qPCR.

**Mouse models**
All wild-type female C57BL/6 mice used in this study were purchased from Beijing Vital River Laboratory Animal Technology and all the experiments in this study were approved by the Scientific Ethic Committee of Huazhong Agricultural University (NO. HZAUMO-2019-019)

and maintained at the Laboratory Animal Centre of Huazhong Agriculture University under specific pathogen-free (SPF) conditions with 12-h light/dark cycles. Room temperature was maintained at 25 °C. The humidity level was controlled between 40 and 60%. Based on the principles of laboratory animal welfare and ethics, this study optimized the design of the project and strictly plans the number of animals required. A total of 28 6-week-old female C57BL/6 mice (20 ± 2 g) were planned. Among them, four were used for bone marrow macrophage isolation experiments, and the remaining 24 were used for Mycobacterium tuberculosis H37Ra infection test and AdoCbl drug treatment experiment.

## Animal experiment

For *M.tb* infection, 6-week-old female mice (20 ± 2 g) were infected by intravenous injection with a dose of $5 \times 10^6$ cfu/mL of 200 μL *M.tb* H37Ra suspension. All animals were randomly distributed into two groups of 12 each. Then mice received a gavage feeding of AdoCbl (Sigma, Cat# C0884) at a daily dose of 0.5 mg/kg·bw, and an equal volume of sterile PBS used as a negative control. These mice were given the therapy for 1-day post-*M.tb* infection. Mice were euthanized after 15 days of drug treatment. Lungs and spleen were taken out for histopathological observation and CFU analysis. A part of the left lung was fixed in 4% paraformaldehyde for a minimum time of 48 h, and tissues were paraffin-embedded to make a section for hematoxylin and eosin (H&E) staining. Homogenates of lungs and spleen were plated for bacterial burden enumeration using a serial dilution method on plates of Middlebrook 7H11 agar containing OADC enrichment and BBL MGIT PENTA antibiotics (BD, Cat# 245114) for inhibition of contamination. CFUs were counted after 3–4 weeks of incubation at 37 °C.

## RNA-seq analysis

FastQC (version: v0.11.8) was used to assess the quality of RNA-Seq reads, and Trimmomatic[54] (version: 0.33) was used to filter out the low-quality bases and adapter sequences. Clean reads longer than 36 bp at both ends were kept for further processing. TopHat[55] (version: v2.1.1) with bowtie2[56] (version: 2.3.5.1) was used to align the paired-end RNA-Seq reads to the human reference genome (hg19) with transcriptome annotations from Ensembl[57]. HTSeq[58] (version: 0.11.2) was used to count the mapped reads on the transcripts corresponding to each gene. DESeq2 (version: 1.36.0)[59] was used for normalization and differential expression analysis with the read counts on the genes as inputs.

## ATAC-seq analysis

The reads filtering steps were the same as in "RNA-Seq analysis". Trimmed reads were aligned to the human genome assembly (hg19) with the Burrows-Wheeler Aligner-MEM[60] (version: 0.7.17-r1188). For each sample, SAMtools[61] (version: 1.7) was used to sort the mapped reads by position. After marking the duplication reads by Picard toolkit (version: 1.119), uniquely mapped reads and non-redundant sequences were kept for further analysis by SAMtools with the parameter F: 1024, q 20. F-Seq[62] (version: 3) was used to call ATAC-Seq peaks as ENCODE ATAC-Seq data analysis pipeline, and the top 100,000 accessible regions for all samples were kept further analysis.

## ChIP-seq analysis

The reads filtering steps were the same as "RNA-Seq analysis". Burrows-Wheeler Aligner-MEM[60] (version: 0.7.17-r1188) was applied to map the reads to the human genome assembly (hg19). Mapped reads were sorted by position, and duplications were discarded by SAMtools[61] (version: 1.7). The reads with a mapping quality higher than 30 were considered uniquely mapped reads. Peaks were called by MACS2[63] (version: 2.1.1.20160309). For broad peaks, the parameters were -B–broad -q 0.05. For narrow peaks, the parameter was -B.

## Annotation of chromatin states of enhancer and promoter

Promoter regions were defined as the genomic regions 1 kb in front of the transcription start sites (TSS) of RefSeq genes. Repressed promoters were defined as promoter regions without H3K4me3 peaks. Bivalent promoters were defined as the promoter regions enriched with H3K4me3 and H3K27me3 peaks. Active promoters were defined as promoter regions enriched with H3K4me3 peaks but without H3K27me3 peaks.

For each cell type, the open non-promoter regions with ATAC-Seq peaks were initially selected as the candidate regions of enhancers. As previously described[16,34], the candidate regions with H3K4me1 peaks were defined as enhancers. The active mark H3K27ac and inactive mark H3K27me3 were combined to define the different states of enhancers. An enhancer with H3K27me3 peaks was defined as a poised enhancer. If there were H3K27ac peaks located in the enhancer region, the enhancer was defined as an active enhancer. The enhancers with neither H3H27ac nor H3K27me3 peaks were considered a primed state.

## In situ DLO Hi-C analysis

As the read1 sequences in the 2 × 150 bp paired-end reads in the in situ DLO Hi-C library contain all the chromatin interaction pair information, only read1 sequences were retained for further analysis. Linker filtering was conducted with DLO Hi-C Tool (version: 0.3.9)[64]. Reads with mapping scores of greater than 32 were retained for subsequent analysis. Via the linker sequence, the sequences with interaction pair information were extracted from the raw reads.

To increase the alignment rate, the restriction endonuclease recognition site was complemented at the end of the sequence. Burrows-Wheeler Aligner-ALN[60] (version: 0.7.17-r1188) was used to align the interaction sequences to the human reference genome (hg19) with the parameter -n 0. Only the uniquely mapped reads with mapping quality (MAPQ) scores of ≥20 were retained and paired for further analysis.

The reference genome was divided into fragments according to the restriction enzyme sites, and the uniquely mapped sequence pairs were aligned to the restriction enzyme fragments. If two ends from a paired-read were mapped to the same restriction enzyme fragment, the paired-read was considered a self-ligation product. If two ends from a paired-read were mapped to two adjacent restriction enzyme fragments, the paired-read was considered a religation product. Both the self-ligation and religation reads were excluded for further analysis. If multiple sequences had both ends aligned to the same positions, only one sequence was retained for further analysis, since such reads probably resulted from PCR amplification.

To identify chromatin loops, interaction matrices were converted to.hic files by the pre command of Juicer Tools (version: 1.9.9)[65] with default resolutions. The HiCCUPS algorithm of Juicer Tools (version: 1.9.9) was used to generate loop lists at resolutions of 5 kb, 10 kb and 25 kb.

## Single-cell DLO Hi-C data analysis

The processing of the sciDLO Hi-C sequencing data is basically as same as the bulk DLO Hi-C data processing[64], with modifications in the PETs extraction step to recognize the barcodes and an additional step to split the final valid reads according to the barcodes. These two additional steps were implemented in the sciDLO Hi-C tools[66] (https://github.com/GangCaoLab/sciDLO, version 0.0.1).

In PETs extraction step, the parameter "−fq1 [R1_file]−fq2 [R2-file] −linker GTCGGANNNNNNNNGCTAGCNNNNNNNNNTCCGAC−enzyme T^TA^A" were used. For the cell split step, the default parameters were used: 1, 2, 4 differences in the hamming distance were allowed for the comparison between "read barcode VS library barcode", "barcode1 VS barcode2 (within same reads)", "barcode R1 VS barcode R2 (barcode within R1 and R2)".

Single-cell embedding using the scHiCTools package[42] (version: 0.0.3), all cells matrix was generated with 1 Mb resolution. For cell embedding using the "InnerProduct" to calculate the similarity between all cells, and using MDS method to mapping all cell's contact matrix to a 2-dimensional vector. Thp-*M.tb* cells are classified into "immune highly activated" and "immune lowly activated" by the threshold of the mean value of all Thp-*M.tb* cell's "separation score", which is equal to the mean distance to the closest 4 Thp1-macro cells in the 2-dimensional space.

To calculate local DoD from single-cell data, the bulk cell DoD calculation pipeline was used with a few modifications including: (1) merged single-cell data as input; (2) the "significant interaction points" in the first step were replaced by the single-cell long-range (>20 kb) contacts; (3) the mean distance was divided by 5 kb, to scale at the same level as the bulk cell DoD value; (4) sampling the same number of the contacts in the compared regions of different samples when DoD comparison between samples was performed.

## Simulation of chromosome three-dimensional structure of single cells

The chromatin was simulated to generate the chromatin 3D structure using software "nuc_dynamic" (version: 1.3.0)[67]. For global chromosomes simulation, the software was run with the parameter: "-s 8.0 4.0 2.0 1.0 0.4 0.2 0.1". For local region structure simulation, the parameter "-s 0.1 0.05 0.01 0.005" was used. And the results were saved as Protein Data Bank (PDB) file format and visualized with PyMOL software (version: 2.3.0).

## Hi-C loop analysis

The Thp1-mono and Thp1-macro contact matrices were randomly downsampled to 351,514,529 intrachromosomal contacts (the same number of intrachromosomal contacts as the Thp1-*M.tb* matrix). The HiCCUPS algorithm was employed to call loops from the normalized Thp1-mono and Thp1-macro libraries. To find the differential interaction loops between two samples, the loops were divided into overlapping and nonoverlapping loops. For differential analysis of overlapping loops, the surrounding $5 \times 5$ window was compared between the two matrices by Wilcoxon tests, as previously described[68]. Loops with *p*-values of less than 0.05 were considered significantly different. For nonoverlapping loops, the fold enrichment values of the peak overall local neighborhoods were also calculated in the sample without the loop, as previously described[69]. If all calculated fold enrichment values of the pixels in the sample without the loop were lower than 1.3, the peak in the sample with the loop was considered a differential peak.

## TAD boundary calling

TAD boundaries were called at a resolution of 40 kb, as previously described[70]. The interaction frequencies within 2 Mb downstream and 2 Mb upstream were compared with the default parameters of DomainCaller (version: 0.1.0). The directionality index (DI) was used to quantify the bias in each bin. Domains were inferred from hidden Markov model (HMM) state calls.

If the distance between two TAD boundaries in two samples was within 80 kb, these TAD boundaries were considered conserved TAD boundaries in the two samples. If the distance between two TAD boundaries was greater than 80 kb, the two TADs were considered shifted TADs[71]. If one TAD in the previous cell state corresponded to multiple smaller TADs in the sample of the subsequent cell state, the TAD was considered a separated TAD. If multiple small TADs in the previous cell state corresponded to a large TAD in the subsequent cell state, this TAD was considered a fused TAD.

The Hi-C datasets were applied to distinguish A and B compartments as previously described[72]. If the values in the first eigenvector were higher than 0, the corresponding bins were marked as A compartments. If the values in the first eigenvector were smaller than 0, the corresponding bins were marked as B compartments.

## TAD "Degree of Disorder" (DoD) and Co-Regulation Score (CRS) calculation

MDkNN[73] pipeline (https://github.com/GangCaoLab/MDkNN, version 0.0.1) was used for TAD DoD calculation. The significant chromatin interactions were first identified based on a Poisson process model. The expected value of the contact matrix was calculated by considering both distance-dependent decay and the local interaction background, as described in the previous studies[39,69]. The window parameters $p$ and $w$ of the donut filter were set to 4 and 7, respectively, according to the previous studies[39,69]. After allocating all significant interactions in the TAD contact matrix $M$, the k nearest neighbors $Pk_i$ with k = 3 for each significant interaction $p_i = (x_i, y_i)$, $i \in [1,n]$ were calculated. The mean distance within the contact matrix ($MD_i$) between the significant point $p_i$ was measured as the mean distance to its k nearest neighbors, as shown in Supplementary Fig. 2f:

$$MD_i = \frac{\sum_j \text{dist}\left(p_i, p_j\right)}{k} \tag{1}$$

Where $\text{dist}(p_i, p_j) = \sqrt{(x_j - x_i)^2 + (y_j - y_i)^2}$ is the Euclidean distance in the matrix heatmap between $p_i$ and $p_j$. The KDTree data structure was used to accelerate the nearest neighbors searching process[74]. The mean value of all local $MDs$, $\text{DoD} = \sum MD_i/n$, in a TAD was defined as the TAD DoD, which represented the "disorder state" of the chromatin interactions within the TAD.

Since the two-sample Kolmogorov–Smirnov (KS) test is sensitive to differences in both the location and the shape of the distribution, it was used for statistical comparisons between the TAD DoD values of two biological samples. Here, the D statistic was calculated as follows:

$$D_{n,m} = \sup_x |F_{1,n}(x) - F_{2,m}(x)| \tag{2}$$

where sup is the supremum function and $F_{1,n}$ and $F_{2,m}$ are the cumulative distribution functions of the $MD$ of different samples. For balancing the effect of sequencing depth, all sample's contact matrices are reconstructed using the same number of valid reads.

The consistency of the overall transcriptional change direction (upregulation or downregulation) of the genes within a TAD during differentiation and activation was defined as the TAD gene co-regulation score (CRS). The regulation direction score (DS) of a TAD t was defined as follows:

$$DS_t = \frac{\sum_{i \in t} \text{sgn}\left(\log_2(\text{FoldChange}_i)\right)}{N_t} \tag{3}$$

where sgn is the sign function, and i is for gene i inside the specific TAD. The TAD CRS was defined as the absolute value of the DS: $CRS_t = \text{abs}(DS_t)$.

## Collection of GWAS-associated risk SNPs

In this study, we collected SNPs from: (1) previous TB-related GWAS studies[21–28] (Supplementary Data 5-subTable1); (2) GWAS catalog (https://www.ebi.ac.uk/gwas); (3) UK Biobank GWAS datasets (http://www.nealelab.is/uk-biobank). For the SNPs from previous TB-related GWAS studies, we kept SNPs with P-values ≤0.0001. After the data merge, all the SNPs were overlapped with our DLO Hi-C loops. Only the SNPs located in the loop anchor regions were kept (Supplementary Data 5-subTable2). The p-values for meta-analyses were not adjusted because the raw GWAS data of several previous GWAS analyses were not available.

## Integrated analysis of GWAS, eQTL, and Hi-C data

Significant eQTLs associated with risk SNPs were obtained from the GTEx portal (http://www.gtexportal.org), which includes gene expression data for 48 different tissues[75]. Chromatin interaction loops with risk SNPs overlapped with significant eQTL gene regions within relevant tissues were selected for further investigation and are listed in Supplementary Data 5-subTable7. The R package ggbio was employed to demonstrate the interactions between SNPs and significant gene regions with Hi-C loops[76].

## Reporting summary

Further information on research design is available in the Nature Research Reporting Summary linked to this article.

## Data availability

The data that support this study are available from the corresponding authors upon reasonable request. All sequencing data generated in this study have been deposited in the Gene Expression Omnibus (GEO) under accession "GSE208046". The ChIP-seq data of NF-κB (GM15510 cell line) are download from ENCOCE Data Coordination Center (ENCODE DCC) (http://hgdownload.cse.ucsc.edu/goldenPath/hg19/encodeDCC/wgEncodeSydhTfbs). The ChIP-seq data of MED1 and BRD4 are download from GEO under accession "GSE208046 [https://www.ncbi.nlm.nih.gov/geo/query/acc.cgi?acc=GSE160670]". The GWAS SNP was collected from the GWAS catalog [https://www.ebi.ac.uk/gwas] and UK Biobank GWAS datasets (http://www.nealelab.is/uk-biobank). Source data are provided with this paper.

## Code availability

Source code of "MDkNN"[73] pipeline used for TAD DoD calculation can be found on GitHub: https://github.com/GangCaoLab/MDkNN. Source code of "sciDLO Hi-C tools"[66] pipeline used for single-cell DLO Hi-C analysis can be found on GitHub: https://github.com/GangCaoLab/sciDLO.

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

## Acknowledgements

This work was supported by the National Natural Science Foundation of China (grants U21A20259, 31872470, and 32221005 to G.C., 31970590 and 31771402 to G.L., 31900432 to D.L., and 31960136 to Z.T.), the National Key Research and Development Project of China (grant 2021YFD1800401 to G.C., and 2021YFC2701201 to G.L.), Fundamental Research Funds for the Central Universities (grants 2662021PY005 to G.L., and 2662018PY025 and 2662019YJ004 to G.C.), National Postdoctoral Program for Innovative Talents (grant BX20180113 to D.L.), China Postdoctoral Science Foundation (grant 2018M640710 to D.L.).

## Author contributions

G.C., G.L., and D.L. contributed with conception of the project and experiment design. D.L., Z.Z., S.Z., L.X., Q.X., J.W., Y.H., X.W., C.C., and Z.C. conducted the in situ DLO Hi-C, sciDLO Hi-C, ChIP-Seq, ATAC-Seq, RNA-seq, Immunofluorescence, DNA-FISH experiments, and generated data. D.L., B.Y., W.Z., Z.H., and R.Y. conducted CRISPR knocked out

experiments. W.X., P.H., D.L., C. Wu, C. Wang, J.M., and X.H. performed data analysis and interpretation. X. Cheng, X. Cao, and A.R. conducted virulent tuberculosis strain H37Rv infection experiment. D.L., G.C., G.L., W.X., P.H., and C. Wu wrote the manuscript, with input from all other authors. R.Z., Z.T., and Z.F. revised the manuscript. G.C. and G.L. supervise the project.

## Competing interests

The authors declare no competing interests.

## Additional information

Da Lin [1,2,3,16], Weize Xu[1,2,16], Ping Hong[4,5,6,16], Chengchao Wu [1,2], Zhihui Zhang[1,2], Siheng Zhang[1,2], Lingyu Xing[1,2], Bing Yang[1,2], Wei Zhou[1,2], Qin Xiao[1,3], Jinyue Wang[1,3], Cong Wang[1,2], Yu He[1,2], Xi Chen[1,2], Xiaojian Cao[1,2], Jiangwei Man[6], Aikebaier Reheman[2,7], Xiaofeng Wu[1,2], Xingjie Hao[8], Zhe Hu[1], Chunli Chen [9,10], Zimeng Cao[1,2,3,11], Rong Yin[12], Zhen F. Fu [13], Rong Zhou[14], Zhaowei Teng [15], Guoliang Li [4,5,6] ✉ & Gang Cao [1,2,3] ✉

[1]State Key Laboratory of Agricultural Microbiology, Huazhong Agricultural University, Wuhan, China. [2]College of Veterinary Medicine, Huazhong Agricultural University, Wuhan, China. [3]College of Bio-Medicine and Health, Huazhong Agricultural University, Wuhan, China. [4]National Key Laboratory of Crop Genetic Improvement, Huazhong Agricultural University, Wuhan, China. [5]Agricultural Bioinformatics Key Laboratory of Hubei Province, Hubei Engineering Technology Research Center of Agricultural Big Data, 3D Genomics Research Center, Huazhong Agricultural University, Wuhan, China. [6]College of Informatics, Huazhong Agricultural University, Wuhan, China. [7]College of Animal Science and Technology, Tarim University, Alar, China. [8]School of Public Health, Tongji Medical College, Huazhong University of Science and Technology, Wuhan, China. [9]College of Life Science and Technology, Huazhong Agricultural University, Wuhan, China. [10]Key Laboratory of Plant Resource Conservation and Germplasm Innovation in Mountainous Region, Guizhou University, Guiyang, China. [11]College of Animal Sciences, Yangtze River University, Jingzhou, China. [12]Department of Hematology, Zhongnan Hospital of Wuhan University, Wuhan, China. [13]Department of Pathology, College of Veterinary Medicine, University of Georgia, Athens, GA, USA. [14]Department of Reproductive Medicine Center, Zhongnan Hospital of Wuhan University, Wuhan, China. [15]The First People's Hospital of Yunnan Province, Affiliated Hospital of Kunming University of Science and Technology, Kunming, China. [16]These authors contributed equally: Da Lin, Weize Xu, Ping Hong. ✉e-mail: guoliang.li@mail.hzau.edu.cn; gcao@mail.hzau.edu.cn

