## [Peer Review File · Nature Communications]

REVIEWER COMMENTS

Reviewer #1 (Remarks to the Author):

Lin et al have performed a comprehensive omics analysis of monocyte/PMA-generated macrophages, and M.tb-infected macrophages integrating GWAS, eQTL, and Hi-C and other methods to assess the 4D genome and dynamic epigenetic structure of macrophages upon infection with M.tb. With single-cell Hi-C methods, they have successfully detected changes of long-range DNA looping, characterizing topology associated domains (TADs). The analyses were performed with undifferentiated and PMA-differentiated THP-1 cells. The omics approach they have taken is carefully executed, with some novel features, for example the use of an entropy measure to detect TAD "degree of disorder" (DoD) as an indicator of translational activity of chromosomal regions. However, the expected dynamic aspect of the responses is not known since only a single time point was examined. The results suggest that M.tb-induced reduction in TAD disorder reveals multi-gene domains with coordinated enhanced translation. This approach indeed revealed TFs, gene clusters and DNA loops potentially relevant to TB, including NFKB, HERC,IFITs, GBPs, and LLRK. Significantly, an inhibitor of LRRK is shown to prevent inhibition of autophagy, inhibits M.tb colony formation tested with infected bone marrow-derived macrophages, and prevents the induction of lesions in spleen and lung tissues of mice in vivo.

Although a technically sound approach is taken with new information on the 4D genome and epigenetics in M.tb-infected cells, relevance of the data to true infection is uncertain. THP-1 cells are a human tumor cell line (not mentioned in the manuscript) and are distinctly different from the phenotype and immune response of primary human macrophages, esp. pulmonary macrophages. The M1/M2 phenotype spectrum is also not closely representative of primary human cells. Likewise, the work was performed entirely with a derived attenuated strain of M.tb H37Ra (a fact about this strain not mentioned in the manuscript) which is well known to deviate from the behavior of virulent strains. Finally, the in vivo work was performed with the attenuated Ra strain administered IV rather than aerosol, an artificial system. Validation work using more relevant pathogenesis systems for the key findings (perhaps collaboratively with readily available scientists that have access to a BSL3) is warranted including a few time points in these experiments to glean more information on the dynamic aspect of the responses.

Several additional issues that need to be addressed include the following.

1. Culture conditions. Merely incubating the cells in vitro will change gene expression – how long were the cells kept in culture before the experiments? PMA and M.tb exposure was for 8 h – what was the rationale for selecting this single time point? M.tb cells need some time to adjust to the macrophage environment, but inferences are made with regards to M.tb growth dynamics (neglecting uptake and adaptation important at this time point). Are the DoD changes maximal at 8h?
2. Multiple statistics are performed but is not clear in each case what the p values are - adjusted to what.
3. While the LRRK2 results are intriguing, some caution is advised in accepting the notion that the LRRK inhibitor is a solid drug candidate. The DNA looping to a domain harboring the eQTL rs1873613 (apparently also a TB GWAS hit) is well documented, but the effects of rs1873613 on expression are rather small [p value 0.000028 and NES 0.093 (better measure)], with no other SNP in LD within this region detectable (genotyping error?), and the functional assay results in Figure 7 are small. The tissue lesions shown are clear, but there is no information as to how many replicates have been done.
4. Editing is needed throughout the manuscript for grammar and structure. For example: "checkpoint protein PD-L1, which can counteracting T cell-activating signal .." and "NF-κB can directly induces PD-L1 gene transcription.."
5. I have questions about the novelty of some of the findings here. Naively I think that there is some retreading of well-established understanding, albeit in the context of a new technique. It is not overly transparent whether this is the case for some of the data shown. For instance, translocation of NFKB to the nucleus upon infection, and the known targets of NFKB following translocation.

6. Degree of Disorder as a concept is not clearly presented. It is not clear early during the presentation of DoD analysis what high and low DoD represent about the nature of chromatin.
7. Line 235-236, not clear what is meant here.
8. There is some discussion of heterogeneity in specific responses from the single cell 4D data. Is this heterogeneity driven by infected vs. uninfected cells within the same condition? More generally can the question of whether the detected changes are driven by direct responses to infection (in an infected cell), from a bystander effect.
9. In Fig 5A, I believe there are only three data points for each cell stage analyzed. However, a specific trajectory of lower DoD is presented where there is a precipitous drop of DoD over a very tight time window. Is there any actual data to support this model? The x-axis also does not appear to be proportional, yet very specific DoD values are used on the y-axis. It is not clear whether this figure is a cartoon representation vs. a visualization of data.
10. I believe that the paper is presented as an integrated Results + Discussion section; however, there is only a Results section listed, no discussion.

Reviewer #2 (Remarks to the Author):

The manuscript entitled "4D Genome Orchestrating Immune Gene Expression of Macrophages during Differentiation and Infection" by Lin et al used a variety of genomic approaches to describe and characterize the changes in macrophage (THP-1) genome organization in bulk and single cells. The authors demonstrated that Mycobacterium tuberculosis infection leads to shifts in TAD boundaries and appearance of new chromatin contacts and loops, especially at the NF- κ B binding sites around immune related genes. They also show that induction of inflammatory gene expression is reflected by a decreased "degree of disorder" (DoD) within TAD supporting that stochasticity of genome organization could be related to its function. Finally, they also integrated their data with GWAS and eQTL analysis to provide functional characterization of tuberculosis susceptible loci. While the study provides a very comprehensive resource and is a useful addition to the field, the manuscript has some shortcomings that should be addressed.

-The manuscript describes an impressive amount of experiments, data and analysis and it could profit from more focus and shortening.

-Much more repressed promoters and enhancers are created during infection (Figure 1f and S1f). Please elaborate on these results and on their significance as currently the analysis is focused on only activation of immune genes. How does the chromatin organization and DoD change at these sites? In line with this, I would suggest the authors to put the current Figure S2r in the main figure.

-The changes in TADs are very prone to differences in the data analysis. Are the changes described in TAD boundaries (Figure 2b) retained if the resolution of analysis is changed? Can these changes be ranked based on their robustness? In line with this, are DoDs also representing TAD boundary changes? And how do DoDs relate to the loops (Figure 5). Please clarify these interrelations to the reader to bring logical flow to the results.

- Please include also information on the TB-related GWAS studies (refs 16-19) and the amount of SNPs identified from each of them and the p-values for each SNP to Table S7. Why was only suggestive cutoff p-value of $< 1 \times 10^{-5}$ used and how many of them would actually pass the GWAS significance? Do these only represent the lead/sentinel SNPs or also the proxies in high linkage disequilibrium that could be causal as well? It is hard to evaluate the robustness of this analysis with this missing information. Please also present colocalization analysis of the GWAS and the eQTL signals or minimally locus zoom plots showing the signals follow a similar pattern.

-Lines 257-259: "By comparing with other single-cell Hi-C methods, we demonstrated that sciDLO Hi-

C datasets contains the highest proportion of proximity ligation junction reads (Supplementary information, Fig. 3b).” Please provide more in-depth comparison of the methods, why would sciDLO Hi-C outperform the others or is this a matter of sequencing depth, cell numbers etc? How many interactions per cell were seen?

-The data generated in this study is not accessible to reviewers under GSE143984 and GSE159501. Please provide the Reviewer token. Also the ENCODE link is not working on line 859.

-Introduction: Please introduce the concept of M1 and M2 macrophages.

-How many replicates were used in the RNA-Seq analysis? Figure S1a only shows two which seems insufficient for statistical power.

-What is known of the bivalent state changes during macrophages? This information should be introduced in the text.

-Altogether, the supplementary figure legends would need more detailed descriptions. To name few: Figure S1F. What does a primed vs poised enhancer means? Please define in the figure legend. Figure S7a. What are the SNPs highlighted in black.

-Figure 3h and i. Do they represent bulk or pseudobulk of the single cells?

-Figure 3j,k, how many of the ~500 cells per group actually show this interaction? Perhaps present a figure similar to 4b in the supplement for this locus.

-Figure 4b. Please increase resolution.

-Figure 4e. Please provide quantification and statistics.

Reviewer #3 (Remarks to the Author):

In this manuscript, Da Lin et al. utilized bulked and single-cell Hi-C seq technology to investigate the chromatin structure remodeling in a "4D" level during the macrophage polarization induced by Mycobacterium tuberculosis infection. They reported functional loci inside the unchanged TAD regions during THP1 cell differentiation showed a distinct "degree of disorder" correlated with the epigenetic states, gene expression, and chromatin accessibility. Interestingly, these regions fit well in a liquid-liquid phase separation model, explaining the high efficient and correlated gene expression pattern. Later, the authors applied the sequencing data to screen the regulatory elements and SNPs susceptible to TB infections. They identified several potential candidate variants, especially gene LRRK2. Later, the functional validation was performed by using LRRK2 inhibitor both in vitro and in vivo.

Although the massive Hi-C data are in high quality and quantity, and there is no doubt that the dynamic chromatin structure contributes to the immunological phenotypes demonstrated in THP1 cells, the current data can not fully support the conclusion.

First of all, the authors used THP1 cells to evaluate the chromatin structure and immune gene expression of macrophages during differentiation. THP1 is a human leukemia monocytic cell line that cannot reflect the natural macrophages phenotypes. The abnormal chromosome rearrangements in the tumor cell line may disturb the conclusion. Human monocytes obtained from healthy donors and macrophages derived from those monocytes were strongly recommended. Otherwise, murine BMDMs may be another option. THP1 cells may be used to verify the mechanical hypothesis but can not be applied to build up the fundamental database. Furthermore, a similar paper published before (Cell

Immunol. 2020 Sep;355:104148.) also used THP1 cell as a model to illuminate the chromatin dynamics before and after LPS polarization. It's a capture Hi-C seq, and not in a whole genome scale, but some similar observations were reported.

Secondly, in the title and the beginning of the manuscript, the authors emphasized the "4D" genome concept, which means the "time" dimension was considered in the whole system. However, the data set were mainly based on the initial monocytic THP1 cell stage and two terminally polarized macrophage-like THP1 cell phases. I understand that it's hard to monitor the chromatin dynamic changes in a real-time manner. But time points, such as an intermediated stage before the terminally differentiated stage, are needed to illustrate the fourth dimension they insisted on.

A new concept of "Degree of Disorder" (DOD) was proposed to evaluate a set of loci inside the unchanged TADs. I do not quite understand the algorithm to measure the DOD, but the results indicated that those regions are newly established chromatin interactions in the TADs. Since the differentiation of the THP1 cells did not induce dramatic TADs changes, these regions may represent the latent or poised enhancers and (or) super-enhancers, which were connected to a batch of gene promoters. The detailed site-specific epigenetic analysis will draw a clear picture of those regions. Using the term "reduction of DOD" may not be proper in this scenario.

Another important discovery in this manuscript is the comprehensive map of TB susceptibility loci and their long-range chromatin interaction. Data about LRRK2 is intriguing, but the detailed explanation about whether and how those SNPs influence the 3D chromatin structure in THP1 cells is missing. SNPs associated point mutation in THP1 cells constructed by Crisper/Cas9 knock-in system will be helpful to solve those puzzles.

Lastly, whether the chromatin structure varies between human and murine macrophages? Although the mouse model was used to clarify the importance of LRRK2 in TB infections, LRRK2 plays the same role in remodeling the chromatin structure in murine macrophages is needed to confirm at first.

Reviewer #4 (Remarks to the Author):

It was a pleasure to read the manuscript by Lin et al on 4D genome dynamics during the differentiation of monocytes into macrophages and subsequent TB infection. Very extensive experiments and high-quality data generation reveal a fascinating view on this process. The clear writing and well-designed figures allow the reader to easily follow the results and the highlighted examples are valid. A few grammatical / copy-paste errors remained and can be solved easily.

I was asked specifically to review the integrative omics analysis. While the LRRK2 example that result from this analysis is convincing, the overall approach has limitations that should be made clear. The authors simply overlap TB GWAS SNPs, GTEx eQTL data on all available tissues, and their own high-quality loop data. There is no statistics involved, and hence it cannot be called an analysis but should be phrased as a prioritization step or so. Prioritization because any overlap is not strong evidence for a mechanism, but merely hypothesis generating. For LRRK2 the story becomes interesting because of additional information. Also note in this context that the authors took eQTLs from all GTEx tissues. Their own data is beautiful and shows how specific the 4D genome is upon differentiation and activation; overlapping that data eQTL data from any tissue except eQTLs from monocytes and macrophages is not enough. In this context it may be surprising that the tissue closest to monocytes/macrophages, namely whole blood, did not result in any of the loops overlapping with a TB SNP and an eQTL.

All in all, this part of the study should be phrased as a prioritization step instead of a systematic analysis; limitations should be noted (no statistics, no monocyte/macrophage-specific QTLs etc). It is the follow-up work and interpretation that make the prioritized examples interesting.

Point-by-Point Response Letter

We'd like to thank all the reviewers for their critical comments, inspiring suggestions and kind considerations. We performed substantial experiments according to the reviewers' comments and suggestions, which have further strengthened our study. In the revised manuscript, the discussion parts are marked with grey shade, and the changes in the manuscript text file are highlighted as blue.

Reviewer #1

Remarks to the Author:

Lin et al have performed a comprehensive omics analysis of monocyte/PMA-generated macrophages, and M.tb-infected macrophages integrating GWAS, eQTL, and Hi-C and other methods to assess the 4D genome and dynamic epigenetic structure of macrophages upon infection with M.tb. With single-cell Hi-C methods, they have successfully detected changes of long-range DNA looping, characterizing topology associated domains (TADs). The analyses were performed with undifferentiated and PMA-differentiated THP-1 cells. The omics approach they have taken is carefully executed, with some novel features, for example the use of an entropy measure to detect TAD "degree of disorder" (DoD) as an indicator of translational activity of chromosomal regions. However, the expected dynamic aspect of the responses is not known since only a single time point was examined. The results suggest that M.tb-induced reduction in TAD disorder reveals multi-gene domains with coordinated enhanced translation. This approach indeed revealed TFs, gene clusters and DNA loops potentially relevant to TB, including NFkB, HERC, IFITs, GBPs, and LLRK. Significantly, an inhibitor of LRRK is shown to prevent inhibition of autophagy, inhibits M.tb colony formation tested with infected bone marrow-derived macrophages, and prevents the induction of lesions in spleen and lung tissues of mice in vivo.

Response: We appreciate the professional comments and inspiring suggestions, which further improve this work. As described in the following and also shown in the revised manuscript, we provided new data to address all the comments.

Q1: Although a technically sound approach is taken with new information on the 4D genome and epigenetics in M.tb-infected cells, relevance of the data to true infection is uncertain. THP-1 cells are a human tumor cell line (not mentioned in the manuscript) and are distinctly different from the phenotype and immune response of primary human macrophages, esp. pulmonary

macrophages. The M1/M2 phenotype spectrum is also not closely
 representative of primary human cells.

FIG.R1

 **Fig.R1. Gene expression pattern and DoD changes of human monocyte derived**
 **macrophages (hMDMs) after *M.tb* infection. (A)** MA plot for gene differential expression
 analysis during virulent tuberculosis strain H37Rv infection. X axis represents the mean of
 normalized counts, Y axis represents the log₂ fold changes of gene expression level. The
 immunity genes which were significantly upregulated in the Thp1-*M.tb* were marked in the figure.
 **(B)** Overlap of KEGG pathway. **(C) and (D)** KEGG pathway enrichment analysis for the
 differentially expressed genes. **(E)** TAD DoD dynamics of NF-κB target locus after H37Rv
 infection in hMDM cells. P values determined by unpaired one-sided t test. **(F)** Log₂ fold
 changes of TAD DoD value of NF-κB target locus after H37Rv infection in hMDM cells. The
 typical NF-κB target gene were marked in the figure. **(G) and (H)** Comparison of TAD DoD
 value changes around the *GBP* locus in hMDM cells before and after H37Rv infection.

 **Response:** Thanks for this important comment. We agree with the reviewer that THP-
 1 is a tumor cell line. However, THP-1-derived macrophages exhibits a very similar
 phenotype and immune response with primary macrophages, such as human monocyte
 derived macrophages (hMDMs) and has been extensively used as a cell line model to
 mimic macrophage^{1, 2}. Researchers have systematically compared hMDMs and THP-
 1-derived macrophages as *in vitro* models for *M.tb* infection. The results showed that
 the difference of bacterial uptake, viability, cytokine and chemokine mRNA expression,
 and cytokine secretion between hMDMs and THP-1-derived macrophages is very small

³. To further verify that our data in human monocytes, the monocytes were isolated from
peripheral blood and induced into hMDMs by macrophage colony-stimulating factor
(M-CSF). After virulent tuberculosis strain H37Rv infection, we collected the infected
cells (hMDMs-*M.tb*) and constructed RNA-Seq and DLO Hi-C libraries. As shown in
FIG. R1A, immunity genes which were significantly upregulated in the Thp-*M.tb*, such
as *GBP1*, *GBP5*, *IFIT3*, *CCR7*, and *PD-L1* (revised manuscript, Table S2), were also
significantly upregulated in hMDMs-*M.tb* (FIG. R1A).
KEGG pathway analysis showed that the majority of the significantly differentially
expressed genes enriched KEGG pathways were consistent (FIG. R1B), such as NF-
kappa B signaling pathway, TNF signaling pathway, and Jak-STAT signaling pathway,
etc (FIG. R1C and D). Furthermore, we validated the TAD DoD dynamics of NF-κB
target locus after *M.tb* infection in hMDM cells, the TAD DoD value of these regions
in hMDMs-*M.tb* was also significantly decreased (FIG. R1E, $P < 0.0001$) compared to
that of the control group, such as *GBP2*, *GBP5*, *NFKB1*, and *TNFSF15* gene locus
(FIG.R1F). Consistent with THP-1-derived macrophages (revised manuscript, Fig. 4a),
the DoD value of *GBP* gene loci decreased from 2.98 to 1.83 after *M.tb* infection in
hMDMs (FIG.R1G,H). These results suggest that the immunoresponse of THP-1
infection model is highly similar to primary human macrophages.

Fig. 4

**Q2:** Likewise, the work was performed entirely with a derived attenuated strain
 of *M.tb* H37Ra (a fact about this strain not mentioned in the manuscript) which
 is well known to deviate from the behavior of virulent strains.

FIG.R2

**Fig.R2. Comparison of THP-1 gene expression patterns after H37Ra and H37Rv infection.**
 **(A) and (B)** MA plot for gene differential expression analysis during H37Ra and H37Rv infection.
 X axis represents the mean of normalized counts, Y axis represents the log₂ fold changes of
 gene expression level. The main immune response genes were marked in the figure. **(C)**
 Overlap of KEGG pathway after H37Ra and H37Rv infection. **(D)** KEGG pathway enrichment
 analysis for the differentially expressed genes after H37Rv infection.

**Response:** Thanks for the comments. To investigate the differences of the host immune
 responses to the attenuated *M.tb* strain H37Ra and the virulent *M.tb* strain, we infected
 THP-1-derived macrophages with H37Ra and H37Rv (a virulent *M.tb* strain) with the
 same MOI (MOI=20) for 12 h, respectively. As shown in FIG. R2A and B, the main
 immune response genes have the similar expression alteration upon H37Ra and H37Rv
 infection. KEGG pathway analysis on the significantly differentially expressed genes
 showed that 78% (39 out of the top50, sort by p-value) KEGG pathways were consistent
 (FIG. R2C). The KEGG pathways which we focus on in our study were conserved
 between H37Ra and H37Rv infection (FIG. R2D), such as NF-kappa B signaling
 pathway (*GBP* genes, *IFIT* genes, *NFKB1*, etc.), Chemokine signaling pathway (*CCL2*,
 *CCL2*, etc.), and Jak-STAT signaling pathway (*STAT1*, *PD-L1*, etc.). What's more, in
 FIG. R1, we verified that our THP-1 infection cell model is highly similar to the
 hMDMs infected with virulent strain H37Rv.

**Q3:** Finally, the *in vivo* work was performed with the attenuated Ra strain
administered IV rather than aerosol, an artificial system. Validation work using
more relevant pathogenesis systems for the key findings (perhaps
collaboratively with readily available scientists that have access to a BSL3) is
warranted including a few time points in these experiments to glean more
information on the dynamic aspect of the responses.

FIG.R3

**FIG.R3. Infected C57BL/6 mice via nasal drip and CFU assays of *M.tb* in lungs.** P values
were calculated by unpaired one-sided t test.

**Response:** As suggested by the reviewer, we tried our best to perform virulent strain
H37Rv aerosol infection in the BSL3 laboratory with animal facility. Unfortunately, we
didn't find any opportunity for this experiment, as the BSL3 animal laboratories are
overwhelmed by COVID-19 experiments. Alternatively, we infected C57BL/6 mice
with H37Ra via nasal drip to verify the reliability of our *in vivo* results. As shown in
FIG. R3, the numbers of CFU were also significantly decreased after 10 and 30 days,
respectively, under AdoCbl treatment in comparison to the PBS control.

Several additional issues that need to be addressed include the following.

**Q4:** Culture conditions. Merely incubating the cells *in vitro* will change gene
expression – how long were the cells kept in culture before the experiments?

**Response:** THP-1 cells were cultured from frozen stocks and passaged three times (3-
139 day/passage cycle) before the experiments.

**Q5:** PMA and *M.tb* exposure was for 8 h – what was the rationale for selecting
this single time point? *M.tb* cells need some time to adjust to the macrophage
environment, but inferences are made with regards to *M.tb* growth dynamics
(neglecting uptake and adaptation important at this time point).

FIG.R4

**FIG.R4. Establish THP-1 infection model. (A)** Infection efficiency of different MOI at 4 hours
after infection. **(B)** Infection efficiency at different time points under MOI=20.

**Response:** In our infection model, Thp1-macro were infected with *M.tb* at MOI
(multiplicity of infection) 20 for 4 hours. After 4 hours, the infected cells were washed
twice with pre-warmed PBS and incubated with fresh culture medium for another 8
154 hours, so the total infection time is 12 hours.

In order to improve the infection efficiency, a high MOI (MOI=20) was used for *M.tb*
infections to make sure most of the cells were infected. As shown in FIG. R4A, under
MOI=20, 4 hours incubation can achieve 81.8% infection rate. However, a side effect
of high MOI is higher cell death rate. As shown in FIG. R4B, when the infection time
reached 22 hours, 69.5% of the cells died. Therefore, we set the infection time to 12
160 hours, which can ensure both the viability of the cells, infection efficiency, and
161 interaction time between host and bacterial during the experiment.

**Notes:** Infection rates were calculated using H37Ra-RFP, and the ratio of dead cells
were calculated by quantifying genome DNA concentration from surviving cells.

**Q6:** Are the DoD changes maximal at 8h?

FIG. R5

**FIG.R5.** The overall DoD value at 12 hours and 22 hours after *M.tb* infection.

**Response:** Thanks for this important point. We assume the reviewer is referring to 12
178 hours here, as we infected the cell with *M.tb* for 12 hours. To address this question, we
compared the overall TAD DoD values of THP-1-derived macrophages at 12 h and 22
180 h after infection. As shown in FIG. R5, the overall DoD value at 12 h was significantly
decreased ($P=1E-16$) compared to the uninfected control. However, the DoD value
decreased from 12h to 22h is not so significant ($P=5.58E-05$). It's hard to draw the
conclusion that the DoD changes maximal at 12 h, but it can be confirmed that there is
a significant decrease in DoD at 12 h after infection.

**Q7:** Multiple statistics are performed but is not clear in each case what the p
values are - adjusted to what.

**Response:** We apology for our unclear description. In the revised Figures, we added a
horizontal line to indicate which two sets of samples were compared, and added a
description of how the p-value is calculated in Figure legends.

**Q8:** While the LRRK2 results are intriguing, some caution is advised in
accepting the notion that the LRRK inhibitor is a solid drug candidate. The DNA
looping to a domain harboring the eQTL rs1873613 (apparently also a TB
GWAS hit) is well documented, but the effects of rs1873613 on expression are
rather small [p value 0.000028 and NES 0.093 (better measure)], with no other
SNP in LD within this region detectable (genotyping error?), and the functional
assay results in Figure 7 are small.

FIG.R6

Fig.R6. The function of rs1873613. (A) Sanger sequencing result of the chr12: 40552317 locus in the THP-1 control cell line. The sgRNA target is marked with a red line. (B) Validation of “T” to “C” single base pair mutation in chr12:40552317 (rs1873613). To avoid the cleavage of the homologous recombination arms of the DNA donor by Cas9 upon transfection, we also mutated the PAM sequence of the sgRNA target (chr12:40552317, “C” to “T”) in our donor plasmid. (C) Histone modification, chromatin state, and Hi-C loop information of rs1873613 and LRRK2 gene region from the UCSC Genome browser (<http://genome.ucsc.edu>). rs1873613 is located in the enhancer region of LRRK2. (D) ChIP-qPCR validation of the H3K4me3 enrichment on the LRRK2 enhancer region. The amount of immunoprecipitated DNA in each sample is represented as signal relative to the total amount of input chromatin (y-axis). P values determined by two-sided Student’s t-test. The final concentration of IFN- γ is 20 ng/ml. (E) Log2 fold change of LRRK2 mRNA expression in THP-1 control cell line and the Thp1-rs1873613 under IFN- γ stimulation. P values determined by two-sided Student’s t-test. (F) CFU assays of M.tb in THP-1 control and Thp1-rs1873613 cell lines. P values determined by unpaired one-sided t test.

Response: To further explore the function of rs1873613, we introduced a “T” to “C” single base pair mutation in chr12:40552317 using the CRISPR/Cas9 system in THP-1 cell line, and named the mutated cell line as Thp1-rs1873613 (FIG. R6A and B). Based on the ENCODE HMM chromatin state information from UCSC browser (<http://genome-asia.ucsc.edu/cgi-bin/hgGateway>), we found that rs1873613 is located in an enhancer upstream of *LRRK2*. We showed that this enhancer interacts with *LRRK2*

promoter through long-range chromatin loop (FIG. R6C). Therefore, we speculated that
rs1873613 would affect the activity of the enhancer, thereby regulating the expression
of *LRRK2* gene.

Since *LRRK2* is an IFN- γ target gene⁴, we stimulated THP-1 control and Thp1-
rs1873613 with IFN- γ , respectively, and then evaluate the enhancer activity by ChIP-
qPCR. As shown in FIG. R6D, upon IFN- γ stimulation, the THP-rs1873613 had more
significant ($P = 0.0071$) active chromatin mark (H3K4me3) enriched in the enhancer
region compared to that of the THP-1 control. This indicates that the enhancer region
of THP-rs1873613 is more sensitive to IFN- γ stimulation and is more active than THP-
1 control. Moreover, rs1873613 significantly enhanced the inductive effect of IFN- γ on
*LRRK2* expression (FIG. R6E).

Since the expression of *LRRK2* is positively correlated with the intracellular survival
of *M.tb*⁵, we therefore investigated the intracellular proliferation of *M.tb* in THP-1
control and Thp1-rs1873613 cell lines. As shown in FIG. R6F, G, the bacterial load of
Thp1-rs1873613 was significantly higher than that of the THP-1 control group. Taken
together, we verified that rs1873613 can enhanced the *LRRK2* enhancer activity,
thereby upregulating the expression of the *LRRK2* gene, which eventually lead to the
susceptibility to tuberculosis.

**Q9:** The tissue lesions shown are clear, but there is no information as to how
many replicates have been done.

FIG.R7

**Fig.R7.** HE-stained lung and spleen sections in the AdoCbl and control (PBS) groups.
Obvious lesions in the lungs and spleen are highlighted by the dashed lines. Each sample was
replicated three times.

**Response:** We apology for our unclear description. Each tissue lesion has 3 replicates.
The other two replicates were shown in Fig. R7.

**Q10:** Editing is needed throughout the manuscript for grammar and structure.
For example: “checkpoint protein PD-L1, which can counteract T cell-activating
signal.” and “NF-κB can directly induce PD-L1 gene transcription.”

**Response:** As suggested, we extensively revised the grammar and structure of the
manuscript.

**Q11:** I have questions about the novelty of some of the findings here. Naïvely I
think that there is some rereading of well-established understanding, albeit in
the context of a new technique. It is not overly transparent whether this is the
case for some of the data shown. For instance, translocation of NFκB to the
nucleus upon infection, and the known targets of NFκB following translocation.

**Response:** We agree with the reviewer that translocation of NF-κB to the nucleus upon
infection and the targets of NF-κB are indeed well-known. The novelty of our study is
not the translocation of NF-κB and the targets. We tried to elucidate the alteration of
chromatin architecture and epigenetic state around NF-κB target regions upon infection.
We also summarized the main points of our work as follows:

1. We developed a highly efficient single cell Hi-C method and delineated chromatin
folding pattern in individual cells and revealed the heterogeneous response to *M.tb* at
single cell level. Moreover, we delineated the comprehensive 4D genome and dynamic
epigenetic landscapes of macrophage during differentiation and infection with
*Mycobacterium tuberculosis (M.tb)*.

2. We proposed a novel concept of TAD “Degree of Disorder” to measure the entropy
of chromatin architecture inside immune related TADs, which are correlated with
chromatin accessibility, gene expression and co-regulation, and possibly phase
separation during differentiation and activation.

3. The GWAS, Hi-C, eQTL, ChIP-Seq, and ATAC-Seq analysis identified the long-
range target genes of mycobacterial disease susceptible SNPs and identified LRRK2 as
a potential drug target, whose inhibitor AdoCbl has an anti-tuberculosis effect both *in*
*vitro* and *in vivo*.

4. In the revised manuscript, we introduced a “T” to “C” single base pair mutation in
chr12:40552317 (rs1873613) to THP-1 cells, and verified that rs1873613 can enhance
the *LRRK2* enhancer activity, thereby upregulate the expression of the *LRRK2* gene,
and eventually increase the susceptibility to tuberculosis.

**Q12:** Degree of Disorder as a concept is not clearly presented. It is not clear
early during the presentation of DoD analysis what high and low DoD represent
about the nature of chromatin.

**Response:** We apologize for the unclear description. The chromatin contacts in low-DoD
regions are more highly organized and have significantly more specific chromatin

interaction with each other inside of this region. There are more chromatin loops inside
the low DoD regions, mediating cis-element interaction occurs. Moreover, the low DoD
regions were enriched with more active chromatin signals (H3K4me3 and H3K27ac).
This will lead to the genes located in low DoD regions are more likely to be co-regulated
and have relatively high expression levels.

**Q13:** Line 235-236, not clear what is meant here.

**Response:** During the process of *M.tb* infection, the DoD value of some TADs will
increase, and some will decrease. Within TADs with decreased DoD values, there are
more up-regulated genes. However, within TADs with increased DoD values, there are
more down-regulated genes.

**Q14:** There is some discussion of heterogeneity in specific responses from the
single cell 4D data. Is this heterogeneity driven by infected vs. uninfected cells
within the same condition? More generally can the question of whether the
detected changes are driven by direct responses to infection (in an infected
cell), from a bystander effect.

**Response:** Thanks for this important comment. Although our infection model can
achieve an 81.8% infection rate, there still some cells that are not infected with *M.tb*
(FIG. R4A). In addition, during the infection process, the *M.tb* uptake also varies
between individual cells. The combination of these factors leads to the heterogeneity.
We have tried our best to increase the infection rate to 81.8%, therefore the detected
changes were mainly caused by direct responses to infection. However, we cannot
completely exclude the minor bystander effect.

**Q15:** In Fig 5A, I believe there are only three data points for each cell stage
analyzed. However, a specific trajectory of lower DoD is presented where there
is a precipitous drop of DoD over a very tight time window. Is there any actual
data to support this model? The x-axis also does not appear to be proportional,
yet very specific DoD values are used on the y-axis. It is not clear whether this
figure is a cartoon representation vs. a visualization of data.

**Response:** The DoD values used on the y-axis is based on actual data (from FigS2P).
To avoid confusion, we rephrased the legend of Fig. 5A.

**Q16:** I believe that the paper is presented as an integrated Results + Discussion
section; however, there is only a Results section listed, no discussion.

**Response:** We thank the reviewer raising this crucial point. We have integrated the
discussion into the results section. In the revised manuscript, we marked the discussion
part with grey shade.

Reviewer #2 (Remarks to the Author):

The manuscript entitled “4D Genome Orchestrating Immune Gene Expression
of Macrophages during Differentiation and Infection” by Lin et al used a variety
of genomic approaches to describe and characterize the changes in
macrophage (THP-1) genome organization in bulk and single cells. The authors
demonstrated that Mycobacterium tuberculosis infection leads to shifts in TAD
boundaries and appearance of new chromatin contacts and loops, especially
at the NF-kB binding sites around immune related genes. They also show that
induction of inflammatory gene expression is reflected by a decreased “degree
of disorder” (DoD) within TAD supporting that stochasticity of genome
organization could be related to its function. Finally, they also integrated their
data with GWAS and eQTL analysis to provide functional characterization of
tuberculosis susceptible loci. While the study provides a very comprehensive
resource and is a useful addition to the field, the manuscript has some
shortcomings that should be addressed. -The manuscript describes an
impressive amount of experiments, data and analysis and it could profit from
more focus and shortening.

**Response:** We do appreciate the positive comments of this reviewer. We also thank all
the great suggestions the reviewer raised.

**Q1:** Much more repressed promoters and enhancers are created during
infection (Figure 1f and S1f). Please elaborate on these results and on their
significance as currently the analysis is focused on only activation of immune
genes. How does the chromatin organization and DoD change at these sites?

FIG.R8

**Fig.R8. Deciphering the function of the genes with new repressed promoters. (A)** DoD
value changes in the TAD with new repressed promoters upon infection. **(B)** KEGG pathway
enrichment analysis for the genes with new repressed promoters. **(C)** MA plot for gene
differential expression analysis of autophagy and apoptosis related genes after infection. X axis
represents the mean of normalized counts, Y axis represents the log₂ fold changes of gene
expression level.

**Response:** Thanks for this important comment. As suggested, we investigated the DoD
changes in the TAD with new repressed promoters upon infection. As shown in Fig.
R8A, the DoD values in these regions were significantly increased ($P < 0.0001$) upon
infection. In addition, the KEGG pathway enrichment analysis showed that these
repressed genes were enriched in mTOR signaling pathway, MAPK signaling pathway,
and PI3K-Akt signaling pathway (FIG. R8B). PI3K/AKT/mTOR pathway is an
intracellular signaling pathway important in regulating the cell cycle, autophagic
process^{6,7} and apoptosis^{8,9}. In many cancers, this pathway is overactive, thus reducing

autophagy and allowing proliferation^{10, 11}. Since the PI3K/Akt/mTOR pathway is
negatively correlated with apoptosis and autophagy, we further investigated the
expression pattern of autophagy and apoptosis related genes during *M.tb* infection (FIG.
R8C). As expected, the gene positively regulating autophagy and apoptosis, such as
*CASP10*, *PIK3R3*, *IL1B*, *HIF1A*, *DDIT4* etc., were significantly up-regulated. However,
the genes negatively regulating autophagy, such as *BCL2* and *MLST8*, were
significantly down-regulated.

**Q2:** In line with this, I would suggest the authors to put the current Figure S2r
in the main figure.

**Response:** As suggested we moved Fig. S2r to the Fig. 2k in the revised manuscript,

**Fig.2k** Numbers of up- and down- regulated genes in the DoD decreased and increased TADs
during *M.tb* infection.

**Q3:** The changes in TADs are very prone to differences in the data analysis.
Are the changes described in TAD boundaries (Figure 2b) retained if the
resolution of analysis is changed? Can these changes be ranked based on their
robustness?

FIG.R9

**Fig.R9. Compare the number of the TADs and intra-TAD genes at different resolutions.**

**(A)** The number of TADs at different resolutions. **(B)** Average TAD size and genes number per

TAD at different resolutions. **(C)** Recaptured times of 40Kb TAD boundary in other resolutions.

X axis represents the TAD boundary numbers at 40Kb resolution. Y axis represents TAD

boundary recaptured times at other resolutions. **(D)** Ranked the TADs based on Log₂ fold

changes of DoD value at 40 Kb resolution during *M.tb* infection.

**Response:** We agree with the reviewer that the changes in TADs are very prone to

differences with different resolutions. Therefore, it is necessary to choose a suitable

resolution to call TAD based on the biological question of interest¹². To better illustrate

this, we re-called TAD with resolutions of 10 Kb, 25 Kb, 40 Kb, and 100 Kb,

respectively. As shown in FIG.R9A, the number of TADs varies at different resolutions,

ranging from 1238 to 22527. When the resolution is 10 Kb, the average TAD size is

just 0.14 Mb, and there is only one gene per TAD on average (FIG. R9B). However,

when the resolution is 100 kb, the average size of TAD reaches 2.53 Mb, with

approximately 20 genes per TAD (FIG. R9B).

Since the purpose of TAD calling in this study is to further analysis of TAD DoD and

investigate the intra-TAD gene interactions, the TAD size should not be too large or

too small. If the TAD size is too large, the background will mask the real changes.
However, if the TAD size is too small, it is not suitable for exploring the interaction
between genes. Eventually, in this study, we chose a resolution of 40 kb to call TAD.
Under this resolution, the average TAD size is 0.79 Mb, and average 6 genes per TAD,
which is the most suitable for TAD DoD and gene coregulation analysis.

Although the TAD domains vary at different resolutions, the detected TAD boundaries
are relatively conservative. As shown in FIG. R9C, about 87.5% of the TAD boundaries
at 40 Kb resolution can be recaptured at other resolutions at least once. As suggested,
we ranked the TADs based on log₂ fold change of DoD value during M.tb infection at
40 Kb resolution (Fig. R9D).

**Q5:** In line with this, are DoDs also representing TAD boundary changes?

**Response:** We apology for the unclear description for the DoD calculation. DoD value
is defined by calculating the average distance between intra-TAD interaction spots.
When we compare the dynamics of the DoD, we only selected the unaltered TADs.
Therefore, DoDs do not represent TAD boundary changes. So, DoD is not designed to
detect the changes in TAD boundary.

**Q6:** And how do DoDs relate to the loops (Figure 5). Please clarify these
interrelations to the reader to bring logical flow to the results.

Fig. 5

**Fig.5b** The relationship between TAD DoD and chromatin loop during differentiation and
activation. The x-axis represents log₂ fold change of TAD DoD value, and the y-axis represent
the number of loop changes in each TAD.

**Response:** We apology for the unclear description about Figure 5b. As shown in the
revised Figure 5b, the x-axis represents log₂ fold change of DoD values, and the y-axis
represents loop number changes. These data suggest that the decrease of the DoD value
within the TAD is accompanied by an increase in the number of loops.

Q7: Please include also information on the TB-related GWAS studies (refs 16-19) and the amount of SNPs identified from each of them and the p-values for each SNP to Table S7.

Response: As suggested, we have marked the number of SNPs collected from each GWAS study and the corresponding P-value in detail (TableS7-subTable1).

Q8: Why was only suggestive cutoff p-value of $< 1 \times 10^{-5}$ used and how many of them would actually pass the GWAS significance?

Response: In this study, we collected SNPs from: 1) previous TB-related GWAS studies; 2) GWAS catalog; 3) UK Biobank GWAS datasets. For the SNPs from TB-related GWAS studies, we kept all of them and did not consider P-values, as the SNP already passed the GWAS significance. For the SNPs from GWAS catalog and UK Biobank GWAS datasets, we set the cutoff of p-value at 1×10^{-5} . After data merge, all the SNPs were overlapped with our DLO Hi-C loops, and only keep the SNPs located in the loop anchor regions. The vast majority of SNPs will be filtered out in this step. Therefore, we set a relatively loose p-value cutoff to collect all the potential candidates for the future study. After we overlapped these SNPs with the DLO Hi-C loop anchor region, only 599 SNPs were retained (TableS7-subTable2). It would be of great interest to further investigate these targets as potential candidates for anti-tuberculosis therapy.

**Q9:** Do these only represent the lead/sentinel SNPs of also the proxies in high
 linkage disequilibrium that could be causal as well? It is hard to evaluate the
 robustness of this analysis with this missing information. Please also present
 colocalization analysis of the GWAS and the eQTL signals or minimally locus
 zoom plots showing the signals follow a similar pattern.

FIG.R6

 **Fig.R6. The function of rs1873613.** (A) Sanger sequencing result of the chr12: 40552317
 locus in the THP-1 control cell line. The sgRNA target is marked with a red line. (B) Validation
 of “T” to “C” single base pair mutation in chr12:40552317 (rs1873613). To avoid the cleavage
 of the homologous recombination arms of the DNA donor by Cas9 upon transfection, we also
 mutated the PAM sequence of the sgRNA target (chr12:40552317, “C” to “T”) in our donor
 plasmid. (C) Histone modification, chromatin state, and Hi-C loop information of rs1873613 and
 LRRK2 gene region from the UCSC Genome browser (<http://genome.ucsc.edu>). rs1873613 is
 located in the enhancer region of LRRK2. (D) ChIP-qPCR validation of the H3K4me3
 enrichment on the *LRRK2* enhancer region. The amount of immunoprecipitated DNA in each
 sample is represented as signal relative to the total amount of input chromatin (y-axis). P values
 determined by two-sided Student’s t-test. The final concentration of IFN-γ is 20 ng/ml. (E) Log₂
 494 fold change of *LRRK2* mRNA expression in THP-1 control cell line (Control group) and the
 495 Thp1-rs1873613 cell line under IFN-γ stimulation. P values determined by two-sided Student’s
 t-test. (F) CFU assays of *M.tb* in THP-1 control and Thp1-rs1873613 cell lines. P values
 determined by unpaired one-sided t test.

 **Response:** We appreciate this good point reviewer raised. Due to the privacy and ethical
 reasons, most of the GWAS raw data are difficult to access. Therefore, the main content

of Figure 7 is more about the collection and presentation of the published GWAS results.
Due to the lack of raw data, we cannot be performed colocalization analysis of the
GWAS and the eQTL. It is also hard to confirm that whether the lead/sentinel SNPs is
also the proxies in high linkage disequilibrium. Nevertheless, to ensure the reliability
of our key finding about rs1873613, we performed more experiments to verify this
finding. As shown in FIG. R6, we introduced a “T” to “C” single base pair mutation in
chr12:40552317 using the CRISPR/Cas9 system in THP-1 cell line. This data validated
that rs1873613 can indeed enhance the *LRRK2* enhancer activity, upregulate the
expression of the *LRRK2* gene, and eventually lead to susceptibility to tuberculosis (see
Reviewer #1, Q8 for more details).

**Q10:** -Lines 257-259: “By comparing with other single-cell Hi-C methods, we
demonstrated that sciDLO Hi-C datasets contains the highest proportion of
proximity ligation junction reads (Supplementary information, Fig. 3b).” Please
provide more in-depth comparison of the methods, why would sciDLO Hi-C
outperform the others or is this a matter of sequencing depth, cell numbers etc?
How many interactions per cell were seen?

**Response:** The sciDLO Hi-C library construction strategy is very different from other
single-cell Hi-C methods. The throughput of sciDLO Hi-C is higher, and about 450
single-cell genome 3D structures can be captured in one experiment. However, the
disadvantage is that the average sequencing depth of each cell will be relatively low
($\sim 7 \times 10^5$ raw reads per cell). As other single-cell Hi-C uses a single-tube single-cell
strategy, the throughput is low, but the sequencing depth of each cell will be high (range
from 1×10^6 to 7×10^7 raw reads per cell). Different methods can be selected for different
research purposes. Since sciDLO Hi-C can enriched the 80 bp DNA fragments
containing the proximity ligation junction through DNA PAGE gel (Supplementary Fig
3a), which can greatly reduce the sequencing noise caused by multiple displacement
amplification, therefore the sciDLO Hi-C datasets had the highest proportion of
proximity junction reads. In the sciDLO Hi-C dataset, an average of 47,107 interactions
can be reached per cell.

**Q11:** -The data generated in this study is not accessible to reviewers under
GSE143984 and GSE159501. Please provide the Reviewer token. Also the
ENCODE link is not working on line 859.

**Response:** We sincerely apologize that we forgot that the previous GEO link was a
private link and have re-generated the reviewer link:

<https://www.ncbi.nlm.nih.gov/geo/query/acc.cgi?acc=GSE208046>

Reviewer token: idgvmewoxlgfpun

In addition, the ENCODE link:

[https://www.encodeproject.org/documents/0eb389f9-d23d-4053-b25b-](https://www.encodeproject.org/documents/0eb389f9-d23d-4053-b25b-1e2826ee5a86/@@download/attachment/ATACpipelineV7.pdf)
 [1e2826ee5a86/@@download/attachment/ATACpipelineV7.pdf](https://www.encodeproject.org/documents/0eb389f9-d23d-4053-b25b-1e2826ee5a86/@@download/attachment/ATACpipelineV7.pdf)
 is accessible in our browser:

**Q12:** -Introduction: Please introduce the concept of M1 and M2 macrophages.
 **Response:** As suggested, we have added descriptions of M1 and M2 macrophages in
 the revised introduction section.

**Q13:** -How many replicates were used in the RNA-Seq analysis? Figure S1a
 only shows two which seems insufficient for statistical power.

FIG.R10

**FIG.R10 Verify the repeatability of RNA-Seq datasets of THP-1 cell line. (A)** Principal
component analysis of THP-1 RNA-Seq datasets. **(B)** Log2 fold changes, P-value, and Padj
value of the differentially expressed genes which we focus on in this study. **(C)** and **(D)** MA plots
of differential expression analysis during M.tb infection with two RNA-seq replicates **(C)** and
three RNA-seq replicates **(D)**.

**Response:** We agree with the reviewer that the replicates are very important. In this
study, RNA-Seq was repeated twice. Unlike primary cells, THP-1 is a single-cell-
derived cell line and grow as a largely homogeneous cell population, which has
generally much better reproducibility of the experimental results in comparison to
animal and primary cultured cell experiments. To further validate the reproducibility,
we performed principal component analysis, which also support
the excellent reproducibility between our THP-1 RNA-Seq repeats (FIG. R10A). In
addition, the P-value and Padj value of the genes we focus on in this study are also
statistically significant. Most genes have P-value and Padj less than 1×10^{-100} (FIG.
R10B). Therefore, we used two repeats of RNA-Seq in this study.

Nonetheless, to verify the reproducibility of the differentially expressed genes
identified by our 2-replicate RNA-Seq results, we reconstructed the RNA-Seq library
(3 replicates per sample). As shown in Fig. R10 C and D, although the cell batches are
different, the main immune response genes have the similar expression alteration.
Therefore, in this study, two repeats were sufficient for differential gene expression
analysis.

**Q14:** -What is known of the bivalent state changes during macrophages? This
information should be introduced in the text.

**Response:** As suggested, we added a description of bivalent state chromatin in the
revised introduction section.

**Q15:** -Altogether, the supplementary figure legends would need more detailed
descriptions. To name few:
Figure S1F. What does a primed vs poised enhancer means? Please define in
the figure legend.

Supplementary Fig. 1

d

**Response:** “primed vs poised enhancer” means that during differentiation or infection,
the state of the enhancer changes from primed to poised. For a clearer description, we
have added a sub-Figure (Supplementary Fig. 1d) to describe the various enhancer and
promoter states.

**Q16:** Figure S7a. What are the SNPs highlighted in black.

**Response:** We apology for our unclear description of this point. The marked SNPs are
from Table S7-Subtable2. Due to the large number of SNPs, we only selected part of
SNPs for display.

**Q17:** -Figure 3h and i. Do they represent bulk or pseudobulk of the single cells?

**Response:** Figure 3h and i represent bulk cell Hi-C heatmap. In the revised manuscript,
we have revised the figure legend of this picture to make it more clearly.

**Q18:** -Figure 3j,k, how many of the ~500 cells per group actually show this
interaction? Perhaps present a figure similar to 4b in the supplement for this
locus.

**Response:** Thanks for the suggestion. There are 10 Thp-macro single cells and 9 Thp-
*M.tb* single cells that show interaction in this region (marked with dashed lines in the
Figure S3g). As suggested, we added the merged single cell contact matrix in Figure
S3g.

g Merged single cell contact matrix

Chr2:191.56 M -192.14 M

**Q19:** -Figure 4b. Please increase resolution.

**Response:** As suggested, we have improved the image resolution of Figure 4b.

b Merged single cell contact matrix

Chr1:89575000-89800000

**Q20:** -Figure 4e. Please provide quantification and statistics.

**Response:** Quantitative and statistical were already included in the previous Fig 4e, in
the revised Figure legend we have marked it more clearly.

Reviewer #3 (Remarks to the Author):

In this manuscript, Da Lin et al. utilized bulked and single-cell Hi-C seq
technology to investigate the chromatin structure remodeling in a "4D" level
during the macrophage polarization induced by Mycobacterium tuberculosis
infection. They reported functional loci inside the unchanged TAD regions
during THP1 cell differentiation showed a distinct "degree of disorder"
correlated with the epigenetic states, gene expression, and chromatin
accessibility. Interestingly, these regions fit well in a liquid-liquid phase
separation model, explaining the high efficient and correlated gene expression
pattern. Later, the authors applied the sequencing data to screen the regulatory
elements and SNPs susceptible to TB infections. They identified several
potential candidate variants, especially gene LRRK2. Later, the functional
validation was performed by using LRRK2 inhibitor both in vitro and in vivo.

Although the massive Hi-C data are in high quality and quantity, and there is no
doubt that the dynamic chromatin structure contributes to the immunological
phenotypes demonstrated in THP1 cells, the current data can not fully support
the conclusion.

**Q1:** First of all, the authors used THP1 cells to evaluate the chromatin structure
 and immune gene expression of macrophages during differentiation. THP1 is a
 human leukemia monocytic cell line that cannot reflect the natural macrophages
 phenotypes. The abnormal chromosome rearrangements in the tumor cell line
 may disturb the conclusion. Human monocytes obtained from healthy donors
 and macrophages derived from those monocytes were strongly recommended.
 Otherwise, murine BMDMs may be another option. THP1 cells may be used to
 verify the mechanical hypothesis but can not be applied to build up the
 fundamental database.

FIG.R1

**Fig.R1. Gene expression pattern and DoD changes of human monocyte derived**
 **macrophages (hMDMs) after *M.tb* infection. (A)** MA plot for gene differential expression
 analysis during virulent tuberculosis strain H37Rv infection. X axis represents the mean of
 normalized counts, Y axis represents the \log_2 fold changes of gene expression level. The
 immunity genes which were significantly upregulated in the Thp1-*M.tb* were marked in the figure.
 **(B)** Overlap of KEGG pathway. **(C)** and **(D)** KEGG pathway enrichment analysis for the
 differentially expressed genes. **(E)** TAD DoD dynamics of NF- κ B target locus after H37Rv
 infection in hMDM cells. P values determined by unpaired one-sided t test. **(F)** \log_2 fold
 changes of TAD DoD value of NF- κ B target locus after H37Rv infection in hMDM cells. The
 typical NF- κ B target gene were marked in the figure. **(G)** and **(H)** Comparison of TAD DoD value
 changes around the GBP locus in hMDM cells before and after H37Rv infection.

**Response:** Thanks for this important comment. We agree with the reviewer that THP-
 1 is a tumor cell line, yet THP-1-derived macrophages exhibits a very similar phenotype

and immune response to primary macrophages such as human monocyte derived
macrophages (hMDMs)^{1,2}. Previous studies have already systematically compared the
differences between hMDMs and THP-1-derived macrophages as *in vitro* models for
*M.tb* infection. The results showed that there is no significant difference of the bacterial
uptake, viability, cytokine and chemokine mRNA express, and cytokine secretion
between hMDMs and THP-1-derived macrophages after *M.tb* infection³. Moreover,
THP-1 grow as a largely homogeneous cell population and eliminating variability
introduced by genetic differences and ensures the reliability and reproducibility of
experimental results¹³.

To further verify that our data in human primary macrophages, the monocytes were
isolated from peripheral blood and induced into hMDMs by macrophage colony-
stimulating factor (M-CSF). After virulent tuberculosis strain H37Rv infection, we
collected the infected cells (hMDMs-*M.tb*) and constructed RNA-Seq and DLO Hi-C
libraries. As shown in FIG.R1A, immunity genes which were significantly upregulated
in the Thp-*M.tb*, such as *GBP1*, *GBP5*, *IFIT3*, *CCR7*, and *PD-L1* (Revised manuscript,
Table S2), were also significantly upregulated in hMDMs-*M.tb* (FIG. R1A).

KEGG pathway analysis showed that the majority of the significantly differentially
expressed genes enriched KEGG pathways were consistent (FIG. R2C), such as NF-
kappa B signaling pathway, TNF signaling pathway, and Jak-STAT signaling pathway,
etc (FIG. R1C and D). Furthermore, we validated the TAD DoD dynamics of NF-κB
target locus after *M.tb* infection in hMDM cells, the TAD DoD value of these regions
in hMDMs-*M.tb* was also significantly decreased (FIG. R1E, $P < 0.0001$) compared to
that of the control group, such as *GBP2*, *GBP5*, *NFKB1*, and *TNFSF15* gene locus
(FIG.R1F). Consistent with THP-1-derived macrophages (Fig. 4a in the revised
manuscript), the DoD value of GBP gene loci decreased from 2.98 to 1.83 after *M.tb*
infection in hMDMs (FIG.R1G,H). These results suggest that the immunoresponse of
THP-1 infection model is highly similar to primary human macrophages.

**Q2:** Furthermore, a similar paper published before (Cell Immunol. 2020
Sep;355:104148.) also used THP1 cell as a model to illuminate the chromatin
dynamics before and after LPS polarization. It's a capture Hi-C seq, and not in
a whole genome scale, but some similar observations were reported.

**Response:** It is true that “Cell Immunol. 2020” paper also used THP-1 as the cell model
and reported some similar observations. However, there is a big difference between the
scopes of their study and ours. They (Cell Immunol. 2020) captured dynamic spatial
genomic looping events at an early stage of LPS stimulation, and performed analysis
of chromatin loops with GWAS-SNPs of complex traits (e.g., cardiovascular disease).
The main content of our manuscript is to use multi-omic analysis, especially the single
cell Hi-C we developed in this study to systematically delineate the epigenetic

landscape dynamics in macrophages during differentiation and *M.tb* infection and
understand the epigenetic regulation of immune-response. Our GWAS multi-omics
analysis is focused on tuberculosis susceptibility SNPs. We summarized the main points
of our work as follows:

1. We developed a highly efficient single cell Hi-C method and delineated chromatin
folding pattern in individual cells and revealed the heterogenous response to *M.tb* at
single cell level. Moreover, we delineated the dynamic epigenetic landscapes of
macrophage during differentiation and infection with *M.tb*.

2. We proposed a novel concept of TAD “Degree of Disorder” to measure the entropy
of chromatin architecture inside immune related TADs, which are correlated with
chromatin accessibility, gene expression and co-regulation, and possibly phase
separation during differentiation and activation.

3. The GWAS, Hi-C, eQTL, ChIP-Seq, and ATAC-Seq analysis identified the long-
range target genes of tuberculosis susceptible SNPs and identified *LRRK2* as a potential
drug target, whose inhibitor AdoCbl has an anti-tuberculosis effect both *in vitro* and *in*
*vivo*.

4. In the revised manuscript, we introduced a “T” to “C” single base pair mutation in
chr12:40552317 (rs1873613) to THP-1 cells, and verified that rs1873613 can enhance
the *LRRK2* enhancer activity, thereby upregulating the expression of the *LRRK2* gene,
eventually leading to susceptibility to tuberculosis.

**Q3:** Secondly, in the title and the beginning of the manuscript, the authors
emphasized the "4D" genome concept, which means the "time" dimension was
considered in the whole system. However, the data set were mainly based on
the initial monocytic THP1 cell stage and two terminally polarized macrophage-
like THP1 cell phases. I understand that it's hard to monitor the chromatin
dynamic changes in a real-time manner. But time points, such as an
intermediated stage before the terminally differentiated stage, are needed to
illustrate the fourth dimension they insisted on.

**Response:** We agree with the reviewer that it would be ideal to have much more
detailed intermediated stage to establish a fine and solid "4D" genome concept.
However, just as the reviewer mentioned, it's hard to monitor the chromatin dynamic
changes in a real-time manner. We actually planned to simultaneously construct RNA-
Seq, ATAC-Seq, ChIP-Seq, DLO Hi-C, and sciDLO Hi-C sequence libraries at each
time point, but it is very challenging to perfectly synchronize and repeat all the
experiments at different small intermediated stages. Therefore, in this study we only
selected 3 representative time points. To be more explicit, we deleted “4D” in title and
abstract in the revised manuscript.

**Q4:** A new concept of "Degree of Disorder" (DOD) was proposed to evaluate a
 set of loci inside the unchanged TADs. I do not quite understand the algorithm
 to measure the DOD, but the results indicated that those regions are newly
 established chromatin interactions in the TADs. Since the differentiation of the
 THP1 cells did not induce dramatic TADs changes, these regions may
 represent the latent or poised enhancers and (or) super-enhancers, which were
 connected to a batch of gene promoters. The detailed site-specific epigenetic
 analysis will draw a clear picture of those regions. Using the term "reduction of
 DOD" may not be proper in this scenario.

Supplementary Fig. 2

Fig. 5

**Response:** We agree with the reviewer that the DoD unchanged or increased regions
 may represent more latent or poised enhancers and (or) super-enhancers. As shown in
 Supplementary Fig 2j, high DoD TADs indeed enriched more latent or poised
 chromatin signals (H3K27me3 and H3K9me3). However, this does not conflict with
 the term "reduction of DOD" we used in the manuscript. As shown in Supplementary
 Fig. 2n and Fig. 5b, "reduction of DOD" implies the chromatin contacts in low-DoD
 regions are more highly organized and have significantly more specific and uniform
 chromatin interaction with each other inside of this region. There are more chromatin
 loops inside the low DoD regions, mediating cis-element interaction occurs. Gene
 located in the DoD reduction regions are more likely to be co-regulated and have
 relatively high expression levels.

**Q5:** Another important discovery in this manuscript is the comprehensive map
 of TB susceptibility loci and their long-range chromatin interaction. Data about
 LRRK2 is intriguing, but the detailed explanation about whether and how those
 SNPs influence the 3D chromatin structure in THP1 cells is missing. SNPs
 associated point mutation in THP1 cells constructed by Crisper/Cas9 knock-in
 system will be helpful to solve those puzzles.

FIG.R6

**Fig.R6. The function of rs1873613.** (A) Sanger sequencing result of the chr12: 40552317
 locus in the THP-1 control cell line. The sgRNA target is marked with a red line. (B) Validation
 of “T” to “C” single base pair mutation in chr12:40552317 (rs1873613). To avoid the cleavage
 of the homologous recombination arms of the DNA donor by Cas9 upon transfection, we also
 mutated the PAM sequence of the sgRNA target (chr12:40552317, “C” to “T”) in our donor
 plasmid. (C) Histone modification, chromatin state, and Hi-C loop information of rs1873613 and
 LRRK2 gene region from the UCSC Genome browser (<http://genome.ucsc.edu>). rs1873613 is
 located in the enhancer region of LRRK2. (D) ChIP-qPCR validation of the H3K4me3
 enrichment on the *LRRK2* enhancer region. The amount of immunoprecipitated DNA in each
 sample is represented as signal relative to the total amount of input chromatin (y-axis). P values
 determined by two-sided Student’s t-test. The final concentration of IFN- γ is 20 ng/ml. (E) Log₂
 805 fold change of *LRRK2* mRNA expression in THP-1 control cell line (Control group) and the
 806 Thp1-rs1873613 cell line under IFN- γ stimulation. P values determined by two-sided Student’s
 t-test. (F) CFU assays of *M.tb* in THP-1 control and Thp1-rs1873613 cell lines. P values
 determined by unpaired one-sided t test.

**Response:** Thanks for this suggestion. As we discussed in the manuscript (Fig. 7b), the
chromatin remodeling of the *LRRK2* site was probably mainly caused by *M.tb* infection
(The chromatin contact matrix we compared in Fig. 7b is the infected group and non-
infected group). Nevertheless, we think introduce this SNP in the THP-1 cell line is a
great idea to validate the function of rs1873613. As shown in FIG. R6, we introduced a
“T” to “C” single base pair mutation in chr12:40552317 using CRISPR/Cas9 system in
THP-1, and verified that rs1873613 can indeed enhance the *LRRK2* enhancer activity
as indicated by the H3K4me3 ChIP-PCR. It could then upregulate the expression of the
*LRRK2* gene, and eventually lead to the susceptibility to tuberculosis (see Reviewer #1,
Q8 for more details).

**Q6:** Lastly, whether the chromatin structure varies between human and murine
macrophages? Although the mouse model was used to clarify the importance
of *LRRK2* in TB infections, *LRRK2* plays the same role in remodeling the
chromatin structure in murine macrophages is needed to confirm at first.

FIG. R11

**Fig.R11. Comparison of DoD value changes around the *Lrrk2* gene locus in mouse bone
marrow-derived macrophages (BMDMs) before and after *M.tb* infection.**

**Response:** Thanks for this important comment. As suggested, we infected the mouse
bone marrow-derived macrophages (BMDMs) with *M.tb* and analyzed the chromatin
structure alteration. As shown in FIG. R11, upon infection, the chromatin structure of
*LRRK2* region was indeed undergoing a similar alteration. The DoD value of the *Lrrk2*
gene region was also decreased after *M.tb* infection, suggestion a conserved epigenetic
regulation mechanism in this region.

Reviewer #4 (Remarks to the Author):

It was a pleasure to read the manuscript by Lin et al on 4D genome dynamics
during the differentiation of monocytes into macrophages and subsequent TB
infection. Very extensive experiments and high-quality data generation reveal
a fascinating view on this process. The clear writing and well-designed figures
allow the reader to easily follow the results and the highlighted examples are
valid. A few grammatical / copy-paste errors remained and can be solved easily.
I was asked specifically to review the integrative omics analysis. While the
LRRK2 example that result from this analysis is convincing, the overall
approach has limitations that should be made clear. The authors simply overlap
TB GWAS SNPs, GTEx eQTL data on all available tissues, and their own high-
quality loop data. There is no statistics involved, and hence it cannot be called
an analysis but should be phrased as a prioritization step or so. Prioritization
because any overlap is not strong evidence for a mechanism, but merely
hypothesis generating. For LRRK2 the story becomes interesting because of
additional information. Also note in this context that the authors took eQTLs
from all GTEx tissues. Their own data is beautiful and shows how specific the
4D genome is upon differentiation and activation; overlapping that data eQTL
data from any tissue except eQTLs from monocytes and macrophages is not
enough. In this context it may be surprising that the tissue closest to
monocytes/macrophages, namely whole blood, did not result in any of the loops
overlapping with a TB SNP and an eQTL.

All in all, this part of the study should be phrased as a prioritization step instead
of a systematic analysis; limitations should be noted (no statistics, no
monocyte/macrophage-specific QTLs etc). It is the follow-up work and
interpretation that make the prioritized examples interesting.

FIG.R6

Fig.R6. The function of rs1873613. (A) Sanger sequencing result of the chr12: 40552317 locus in the THP-1 control cell line. The sgRNA target is marked with a red line. (B) Validation of “T” to “C” single base pair mutation in chr12:40552317 (rs1873613). To avoid the cleavage of the homologous recombination arms of the DNA donor by Cas9 upon transfection, we also mutated the PAM sequence of the sgRNA target (chr12:40552317, “C” to “T”) in our donor plasmid. (C) Histone modification, chromatin state, and Hi-C loop information of rs1873613 and LRRK2 gene region from the UCSC Genome browser (<http://genome.ucsc.edu>). rs1873613 is located in the enhancer region of LRRK2. (D) ChIP-qPCR validation of the H3K4me3 enrichment on the *LRRK2* enhancer region. The amount of immunoprecipitated DNA in each sample is represented as signal relative to the total amount of input chromatin (y-axis). P values determined by two-sided Student’s t-test. The final concentration of IFN- γ is 20 ng/ml. (E) Log₂ fold change of *LRRK2* mRNA expression in THP-1 control cell line (Control group) and the Thp1-rs1873613 cell line under IFN- γ stimulation. P values determined by two-sided Student’s t-test. (F) CFU assays of *M.tb* in THP-1 control and Thp1-rs1873613 cell lines. P values determined by unpaired one-sided t test.

Response: Thanks for the comments and suggestions. Due to the privacy and ethical reasons, most of the GWAS raw data are not available, therefore, the main content of previous Figure 7 is more about the collection and presentation of published GWAS results. We agree with the reviewer that without deep integrated analysis with all the raw data, it is inappropriate to call this as real systematic integrative omics analysis. Thus, we rephrased this part, deleted the “integrative omics analysis” words, and

890 discussed about the limitations (no statistics, no monocyte/macrophage-specific QTLs
etc) in the revised manuscript. Moreover, to further validate the reliability of our key
finding of this prioritization step, rs1873613 SNP for susceptibility to tuberculosis, we
introduced a “T” to “C” single base pair mutation in chr12:40552317 by CRISPR/Cas9
system in THP-1 line (FIG. R6). Our data demonstrated that rs1873613 can enhance
the *LRRK2* enhancer activity as indicated by the H3K4me3 ChIP-PCR. It could then
upregulate the expression of the *LRRK2* gene, and eventually lead to the susceptibility
to tuberculosis (see Reviewer #1, Q8 for more details).

**References:**

- 1. Genin, M., Clement, F., Fattaccioli, A., Raes, M. & Michiels, C. M1 and M2
macrophages derived from THP-1 cells differentially modulate the response of
cancer cells to etoposide. *BMC cancer* **15**, 1-14 (2015).
- 2. Lund, M.E., To, J., O'Brien, B.A. & Donnelly, S. The choice of phorbol 12-
myristate 13-acetate differentiation protocol influences the response of THP-
1 macrophages to a pro-inflammatory stimulus. *Journal of immunological methods*
**430**, 64-70 (2016).
- 3. Madhvi, A., Mishra, H., Leisching, G., Mahlobo, P. & Baker, B. Comparison of
human monocyte derived macrophages and THP1-like macrophages as in vitro
models for M. tuberculosis infection. *Comparative immunology, microbiology*
*and infectious diseases* **67**, 101355 (2019).
- 4. Gardet, A. et al. LRRK2 is involved in the IFN- γ response and host response
to pathogens. *The Journal of Immunology* **185**, 5577-5585 (2010).
- 5. Härtlova, A. et al. LRRK2 is a negative regulator of Mycobacterium
tuberculosis phagosome maturation in macrophages. *The EMBO journal* **37**, e98694
(2018).
- 6. Heras-Sandoval, D., Pérez-Rojas, J.M., Hernández-Damián, J. & Pedraza-
Chaverri, J. The role of PI3K/AKT/mTOR pathway in the modulation of autophagy
and the clearance of protein aggregates in neurodegeneration. *Cellular*
*signalling* **26**, 2694-2701 (2014).
- 7. Wu, Y.-T., Tan, H.-L., Huang, Q., Ong, C.-N. & Shen, H.-M. Activation of the
PI3K-Akt-mTOR signaling pathway promotes necrotic cell death via suppression
of autophagy. *Autophagy* **5**, 824-834 (2009).
- 8. Saiki, S. et al. Caffeine induces apoptosis by enhancement of autophagy via
PI3K/Akt/mTOR/p70S6K inhibition. *Autophagy* **7**, 176-187 (2011).
- 9. Wang, D. et al. Leptin regulates proliferation and apoptosis of colorectal
carcinoma through PI3K/Akt/mTOR signalling pathway. *Journal of biosciences*
**37**, 91-101 (2012).
- 10. Daver, N. et al. A phase I/II study of the mTOR inhibitor everolimus in
combination with HyperCVAD chemotherapy in patients with relapsed/refractory
acute lymphoblastic leukemia. *Clinical Cancer Research* **21**, 2704-2714 (2015).
- 11. Franke, T.F., Hornik, C.P., Segev, L., Shostak, G.A. & Sugimoto, C. PI3K/Akt
and apoptosis: size matters. *Oncogene* **22**, 8983-8998 (2003).
- 12. Zufferey, M., Tavernari, D., Oricchio, E. & Ciriello, G. Comparison of
computational methods for the identification of topologically associating
domains. *Genome biology* **19**, 1-18 (2018).
- 13. Phanstiel, D.H. et al. Static and dynamic DNA loops form AP-1-bound activation
hubs during macrophage development. *Molecular cell* **67**, 1037-1048. e1036 (2017).

REVIEWERS' COMMENTS

Reviewer #1 (Remarks to the Author):

The authors have added in all the additional clarifications and information which is needed (in my view) for publication. I do not think further revisions are needed.

Reviewer #2 (Remarks to the Author):

I appreciate the effort invested by the authors in addressing my comments in Q1-Q7, Q10-Q20 which has significantly improved the manuscript. However, the concerns in Q8 and Q9 were not completely addressed.

-Q8. The authors should remove the GWAS SNPs that do not pass at least the suggestive p-value significance. If I understand correctly, the current list in Table S7 contains also SNPs that do not even pass 0.05. Why are those retained? It is very important to acknowledge that such suggestive p-values by no means support causality. For example, the association p-value for the LRRK2 rs1873613 is only 0,000051.

-Q9. Despite rs1873613 being a cis-eQTL for LRRK2 in lung, this SNP is not the top variant associated with the expression which again questions the causality.

<https://gtexportal.org/home/browseEqtls?location=rs1873613&tissueName=Lung> I would suggest the authors to consider cis-eQTLs from monocyte-macrophages instead of GTEX which should be much more relevant for the cellular model used (see for example <https://www.ebi.ac.uk/eqt/Studies/>)

-The new functional data does now nicely demonstrate that the SNP regulates the expression of LRRK2 but whether this is the causal mechanism for mycobacterial disease susceptibility is not demonstrated and thus the results should be treated with caution.

Reviewer #3 (Remarks to the Author):

Authors have applied substantial experiments and data to fully address all the questions and concerns.

Reviewer #2 (Remarks to the Author):

I appreciate the effort invested by the authors in addressing my comments in Q1-Q7, Q10-Q20 which has significantly improved the manuscript. However, the concerns in Q8 and Q9 were not completely addressed.

Response: We really appreciate the reviewer for the critical comments and inspiring suggestions, which significantly improved our study.

-Q8. The authors should remove the GWAS SNPs that do not pass at least the suggestive p-value significance. If I understand correctly, the current list in Table S7 contains also SNPs that do not even pass 0.05. Why are those retained? It is very important to acknowledge that such suggestive p-values by no means support causality. For example, the association p-value for the *LRRK2* rs1873613 is only 0,000051.

Response: We apology for the unclear description about this point. TableS7-subTable1 is just a collection and presentation of SNPs from previous published TB-related GWAS studies¹⁻⁸. This table simply kept all of SNPs and did not consider P-values. As suggested by the reviewer, we retained only SNPs with P value ≤ 0.0001 in the revised TableS7-subTable1. We also clearly described in the revised manuscript that the GWAS study is for association, not for causality.

-Q9. Despite rs1873613 being a cis-eQTL for *LRRK2* in lung, this SNP is not the top variant associated with the expression which again questions the causality. <https://gtexportal.org/home/browseEqtl?location=rs1873613&tissueName=Lung> I would suggest the authors to consider cis-eQTLs from monocyte-macrophages instead of GTEX which should be much more relevant for the cellular model used (see for example <https://www.ebi.ac.uk/eqtl/Studies/>)

Response: Thanks for this great comment. As suggested, we contacted the Wellcome Trust Sanger Institute (datasharing@sanger.ac.uk), Blueprint Data Access Committee (blueprint-dac@ebi.ac.uk), and Dr. Etienne Patin (etienne.patin@pasteur.fr) immediately to request the raw expression data and raw genotype data for further eQTL analysis. From the summary statistics of macrophage/monocyte models provided by Dr. Patin, we did find more *LRRK2*-related SNPs with more significant P-values. However, none of the SNPs has been previously reported in mycobacterial disease related GWAS studies. We agree with the reviewer that these SNPs may also have regulatory effect on *LRRK2*. However, the current goal of this study is to explore the potential regulation mechanisms of Tuberculosis (TB) GWAS SNPs. TB most commonly occurs in the lungs⁹ (known as pulmonary tuberculosis). rs1873613 is the only previously reported GWAS SNP¹⁻⁸ which also acts as a cis-eQTL for *LRRK2* in lung. Moreover, the single base pair mutation experiment via CRISPR/Cas9, also verified that rs1873613 can indeed upregulate the expression of the *LRRK2* gene (Figure 7). Nevertheless, we agree the reviewer that these experiments can only prove the association, but not causality of rs1873613 to TB susceptibility. It would be of great importance to further investigate the role of rs1873613 and other SNPs in the pathogenesis of TB.

-The new functional data does now nicely demonstrate that the SNP regulates the expression of LRRK2 but whether this is the causal mechanism for mycobacterial disease susceptibility is not demonstrated and thus the results should be treated with caution.

Response: We agree with the reviewer that although different experiments suggest a highly correlation between the regulation of *LRRK2* by rs1873613 and mycobacterial disease susceptibility, rs1873613 cannot be considered as a causal mechanism of TB susceptibility. In the future, more experiments need to be done *in vivo* to further validate the function of rs1873613 and other SNPs in mycobacterial disease susceptibility. In the revised manuscript, we rediscussed the potential correlation of rs1873613 and mycobacterial disease susceptibility.

References:

1. Thyé, T. et al. Genome-wide association analyses identifies a susceptibility locus for tuberculosis on chromosome 18q11.2. *Nat. Genet.* **42**, 739 – 741 (2010).
2. Thyé, T. et al. Common variants at 11p13 are associated with susceptibility to tuberculosis. *Nat. Genet.* **44**, 257 – 259 (2012).
3. Curtis, J. et al. Susceptibility to tuberculosis is associated with variants in the ASAP1 gene encoding a regulator of dendritic cell migration. *Nat. Genet.* **47**, 523 – 527 (2015).
4. Sobota, R.S. et al. A locus at 5q33. 3 confers resistance to tuberculosis in highly susceptible individuals. *The American Journal of Human Genetics* **98**, 514–524 (2016).
5. Chimusa, E.R. et al. Genome-wide association study of ancestry-specific TB risk in the South African Coloured population. *Human molecular genetics* **23**, 796–809 (2014).
6. Zhang, F.-R. et al. Genomewide association study of leprosy. *New England Journal of Medicine* **361**, 2609–2618 (2009).
7. Qi, H. et al. Discovery of susceptibility loci associated with tuberculosis in Han Chinese. *Human Molecular Genetics* **26**, 4752–4763 (2017).
8. Zheng, R. et al. Genome-wide association study identifies two risk loci for tuberculosis in Han Chinese. *Nat. Commun.* **9**, 4072 (2018).
9. Bennett, J.E., Dolin, R. & Blaser, M.J. Mandell, Douglas, and Bennett's principles and practice of infectious diseases E-book. (Elsevier Health Sciences, 2019).